



# Improved modelling of mountain snowpacks with spatially distributed precipitation bias correction derived from historical reanalysis

Manon von Kaenel[1], Steven A. Margulis[1]

[1]Civil and Environmental Engineering, University of California Los Angeles, Los Angeles, 90095, USA

*Correspondence to*: Manon von Kaenel (mvonkaenel@g.ucla.edu)

**Abstract.** Accurate estimates of snow water equivalent (SWE) are essential for effective water management in regions dependent on seasonal snowmelt. However, significant biases and high uncertainty in mountain precipitation data products pose significant challenges. This study leverages a SWE reanalysis framework and historical dataset to derive factors that can

downscale and bias-correct mountain precipitation in a real-time modelling context. We evaluate through hindcast modelling how different versions of this precipitation bias correction affect errors in 1 April SWE estimates within a representative snow-dominated watershed in the Western U.S. We also evaluate how the additional assimilation of fractional snow-covered area (fSCA) or snow depth observations during the accumulation season impact the 1 April SWE estimates. Results show that spatially distributed historically informed precipitation bias correction significantly improves SWE estimates, reducing the

normalized root mean square difference (NRMSD) by 58%, increasing the correlation (R) by 43%, and decreasing mean difference (MD) by 88%. The primary strength of this bias correction method lies in capturing the spatial distribution of precipitation bias rather than its interannual variability. Assimilating snow depth observations further reduces errors both at the watershed scale (NRMSD less by 46%) and pixel level in most years, while accumulation season fSCA assimilation is not generally useful. We demonstrate the value of these methods for streamflow forecasts: bias-corrected precipitation improves

the correlation between daily simulated snowmelt and observed streamflow by 31-39% and reduces bias in predicted April-July runoff volumes by 46-52%. This study highlights how historical SWE reanalysis datasets can be leveraged and applied in a real-time context by informing precipitation bias correction.

## 1 Introduction

Seasonal snowpack is a natural water tower; by storing winter precipitation and releasing it as snowmelt, it provides

an essential resource for downstream ecosystems and an estimated 20% of the Earth's population (Dozier, 2011). In order to make critical management decisions for flood control, hydropower operations, irrigation, and other competing demands in snow-dependent regions of the world, water managers need accurate assessments of the distribution and availability of water in snowpack (e.g., Hamlet et al., 2002; Koster et al., 2010; He et al., 2016). Estimating the spatiotemporal distribution and





change of snow water equivalent (SWE) remains a significant and important challenge for the snow hydrology community

(e.g., Cho et al., 2022; Dozier et al., 2016; Lettenmaier et al., 2015).

Large-scale and temporally continuous SWE (snow water equivalent) measurements are generally absent from the real-time observational record. In situ data from networks such as the Western U.S. SNOTEL (snow telemetry) network are not always representative of the heterogeneity of SWE distribution in topographically complex mountain landscapes (e.g., Herbert et al., 2024), and such networks are sparse globally. For example, although seasonal snowpack is crucial to local water

availability in High Mountain Asia, the region has almost no in situ data (e.g., Liu et al., 2021). Remote sensing can provide measurements of snow properties like fractional snow-covered area (fSCA, e.g., Selkowitz et al., 2017), snow depth (e.g., Painter et al., 2016), or albedo (e.g., Bair et al., 2019) over large areas, but there is currently no reliable way of measuring SWE from spaceborne platforms (Lettenmaier et al., 2015).

This implies a continued need for modelling of mountain SWE. Land surface models are commonly used to estimate

SWE and other hydrologic variables over large spatial extents (Cho et al., 2022; Clark et al., 2011; Kumar et al., 2013), but these are susceptible to uncertainties driven by biases in forcing data or model parameterization (Cho et al., 2022). Uncertainty in precipitation products in mountainous terrain, and its implications for SWE and downstream hydrology modelling, is a widely acknowledged challenge (e.g., Schreiner-McGraw et al., 2020; Cho et al., 2022; Pan et al., 2003; Raleigh et al., 2015; Liu and Margulis, 2019). Fang et al., 2023 found that the uncertainty of SWE estimates from commonly used global and

regional modelling products is primarily explained by precipitation uncertainty. Data assimilation has gained popularity as a way to constrain or correct uncertain model estimates of snow with observations of variables such as fSCA or snow depth, and has demonstrated ability as a method to quantify SWE over both melt and accumulation seasons (Margulis et al., 2016; Cortes et al., 2016; Liu et al., 2021; Fang et al., 2022). This approach is particularly valuable in regions where in situ data are sparse but remotely sensed observations like fSCA are available, such as High Mountain Asia (Liu et al., 2021) or the South American

Andes (Cortes et al., 2016). However, such products are typically only generated retrospectively. Recent studies have shown promise in combining historical reanalysis snow estimates with in situ and/or remotely sensed snow observations using statistical methods to specifically develop near real-time SWE estimates (Pflug et al., 2022; Schneider and Molotch, 2016; Bair et al., 2018; Zheng et al., 2018; Yang et al., 2022). While these methods still heavily rely on ground SWE observations, they do demonstrate the value of and potential for historical reanalysis SWE datasets to inform SWE estimation in an

operational context.

Real-time spatially distributed SWE estimates have significant potential for application to water management. Climate change impacts in snow-influenced systems, such as earlier runoff of snowmelt and drops in snowpack volume, pose important challenges for water managers (e.g., Berg and Hall, 2017). Accurate and timely seasonal streamflow forecasts help inform management decisions that allocate resources in a way that is resilient to climate variability or drought (e.g., Tanaka et

al., 2006). Ensemble streamflow prediction (ESP) uses hydrologic models to forecast future streamflow from current snow, soil moisture, river and reservoir conditions (Wood et al., 2002). The skill of these model-based streamflow forecasts is primarily derived from initial SWE and soil moisture conditions (Koster et al., 2010). This suggests that accurate spatially





continuous real-time SWE estimates could be used to reduce uncertainty and error in streamflow forecasts in snow-dominated regions.

In this paper we leverage a SWE reanalysis framework and historical dataset to derive mountain precipitation bias correction estimates, and develop and test spatially continuous SWE estimates on 1 April. The motivating questions are: 1) To what extent can historically informed mountain precipitation bias correction improve model-based spatial SWE estimates? 2) How does the assimilation of accumulation season fSCA and snow depth measurements into this framework affect those estimates? 3) How are snowmelt-driven streamflow predictions affected under these scenarios? We validate these methods

over a well-documented study domain, with the potential to extend to areas that have less access to in situ data.

## 2 Methods

### 2.1 Study domain

The study domain comprises the Hetch Hetchy watershed, a headwater catchment for the Tuolumne River in the California Sierra Nevada (Fig. 1). Its drainage area (~1,200 km$^2$) is characterized by complex topography with elevations

ranging from 1,150 m to 3,850 m. It is representative of other snow-dominated catchments that provide key water supply in the Sierra Nevada. More broadly, it is a demonstrative basin that represents global mountain watersheds where the tested methods could provide utility for water management purposes; that is, basins with complex terrain at high elevations and seasonal snowpack that plays a significant role in the water budget.

The Hetch Hetchy reservoir at the watershed's outlet provides water supply for about 2.7 million residents of the San

Francisco Bay Area, primarily from snowmelt. This watershed also includes a unique Airborne Snow Observatory (ASO) snow depth dataset (Painter et al., 2016) which provides multitemporal lidar-derived snow depth measurements per year. A subwatershed that drains through the USGS TGC (Tuolumne River at Grand Canyon) gauge located at the inlet of the reservoir was delineated for the streamflow analysis.





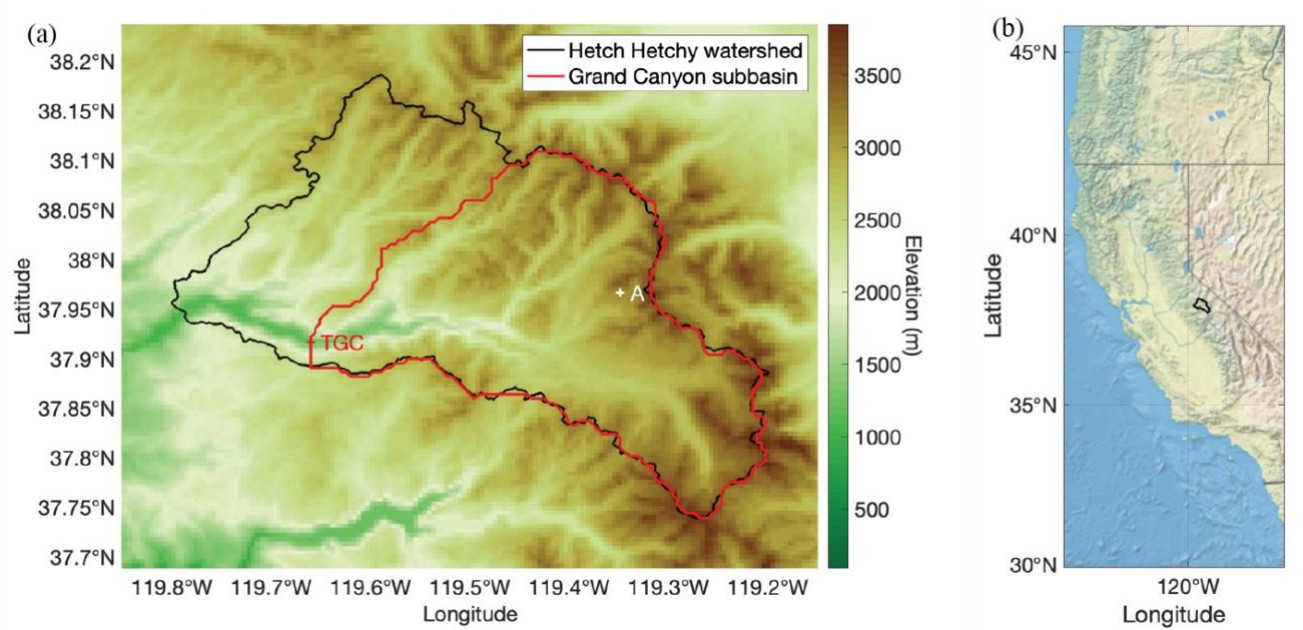

**Figure 1. (a) A map outlining the Hetch Hetchy and the Grand Canyon subwatersheds of the Tuolumne River. The locations of the TGC streamflow gauge and sample model pixel A. (b) Outline of the Hetch Hetchy watershed illustrating its location in the central California Sierra Nevada.**

## 2.2 Overview of SWE reanalysis framework

A Bayesian reanalysis framework (Margulis et al., 2015) is used in this study for both the development of a historical reference SWE dataset, the derivation of historical precipitation bias correction through retrospective analysis, and testing "real-time" applications using that historical data (along with other data). Note that we use a hindcasting approach to model the real-time applications. Details of the application of this framework in a historical context is in section 2.3 and in a real-time context in section 2.4; here, we provide a general overview of the method and its previous applications.

Typically applied retrospectively, this reanalysis framework generates spatiotemporally continuous SWE estimates using a particle batch smoother (PBS) data assimilation technique that constrains a prior ensemble of modeled snow estimates with independent observations (most commonly, satellite-based fSCA measurements). The method was developed by Margulis et al. (2015) and has since demonstrated ability to reproduce observed SWE across global mountain regions: Sierra Nevada (Margulis et al., 2016), South American Andes (Cortes et al., 2016), High Mountain Asia (Liu et al., 2021) and the Western US (Fang et al., 2022). It has also demonstrated success in assimilating remotely-sensed snow depth measurements for SWE estimation (Margulis et al., 2019).

First, an ensemble of prior snow estimates is generated using a forward land surface model; here, the modelling core is the SSiB-SAST LSM (Sun and Xue, 2001; Xue et al., 2003) paired with the Liston Snow Depletion Curve (Liston 2004). This LSM is driven by meteorological forcings which each explicitly incorporate some measure of a priori bias and uncertainty.



Note that implicit in this SWE estimation is the assumption that precipitation (snowfall) in mountainous regions is the largest
source of error in the modelling of SWE. Precipitation fields from raw products like MERRA2 tend be coarse, smooth, and
biased (Liu and Margulis, 2019; Fig. 2a, b). This is acknowledged by the bias correction and large uncertainty in the postulated
prior precipitation distribution, represented by:

$$P_j^-(t) = b_j^- * P_{nom}(t) \tag{1}$$

where $P_j^-(t)$ is the prior precipitation value for ensemble $j$ at time step $t$, $P_{nom}$ is the nominal precipitation estimate (i.e.,
interpolated from MERRA2 as in Fig. 2a), and $b_j^-$ is a scaling factor where the " $-$ " superscript represents that this is a prior
estimate not conditioned on independent observations. The ensemble of scaling factors $b$ (Margulis et al., 2019) are effectively
seasonal multiplicative bias correction factors for precipitation. The prior $b$ values are prescribed as a lognormally distributed
multiplicative factor that describes first-order bias and uncertainty in the nominal precipitation.

Second, a reanalysis step incorporates independent measurements such as fSCA using a Bayesian PBS update. The a
priori (equal) prior weights assigned to each ensemble number are updated to posterior weights that reflect the likelihood that
the ensemble member fits the assimilated measurements. These posterior weights are applied to prior ensemble estimates of
SWE to derive posterior estimates. Note that fSCA and other potential measurements used for assimilation are connected based
on physical processes in the model to other snow variables such as SWE; thus, the whole suite of snow variables is updated
both before and after the assimilated measurement time step. So, although this framework has mainly been used to derive
posterior SWE estimates, a by-product of this is posterior estimates of all snow states/fluxes and variables like $b$ described
above.

## 2.3 Development of historical reference dataset and precipitation bias correction

To generate the historical reference SWE dataset for this study, the SWE reanalysis framework was applied to the
study domain in the same way as in Fang et al., 2022, but with an increased ensemble size and initial conditions set to default
values to focus on the derivation of posterior $b$ values for testing herein. Forcings were sourced from hourly MERRA2 near-
surface meteorological forcing data, and the uncertainty models used to perturb input air temperature, precipitation, dew point
temperature, and shortwave radiation, as well as model parameters, use the values derived for the Western U.S. domain by
Fang et al. (2022) following the methods outlined in Liu and Margulis (2019).

Measurements from Landsat-derived fSCA (raw resolution of ~30 m aggregated to modelling resolution) provide the
data assimilated into the historical reference dataset. We apply screening methods consistent with Fang et al. (2022) to exclude
Landsat observations with cloud cover fraction greater than 40% and individual cloudy pixels with an internal cloud mask. All
remaining fSCA measurements are assimilated into the reanalysis retrospectively and as a batch for each water year. A
measurement error standard deviation for retrieved Landsat fSCA is specified as 10% (Fang et al., 2022). A uniform spatial
resolution of 16 arcseconds (~500 m) and an ensemble size of 100 members is chosen, with hourly outputs aggregated to a
daily timestep for water years (WYs) 1985 to 2021. Initial conditions are set to default values at the start of each water year;





for SWE, that value is 0. This assumes that the seasonal snowpack melts out yearly; although this may not happen every year especially at high-elevation shaded areas of the watershed, we argue this is an assumption worth making to avoid accumulating error and to make fair and consistent comparisons between simulations.

In addition to generating a high-resolution reference dataset of SWE estimates for a 37-year historical period to use for validation, this application of the SWE reanalysis framework also yields a rich database of historical precipitation bias correction factors that are conditioned on assimilated fSCA measurements. These values provide insight into the historical distribution of precipitation bias and uncertainty. Hereafter, and unless otherwise indicated, $b$ will refer to the ensemble mean of the posterior $b$ distribution; this is used interchangeably with "bias correction". This database comprises 37 years * 100 ensemble realizations = 3,700 values of $b$ at each pixel. A historical distribution of $b$ can then be derived at each pixel and for each water year, as demonstrated in Liu and Margulis (2019). This study leverages the insights stored in $b$ towards developing real-time SWE estimates by using them to inform the precipitation bias correction of real-time applications (section 2.4).

Figure 2 illustrates how, for sample water year 2016, the $b$ from the historical reference dataset provides valuable built-in downscaling and bias correction information. Note that the posterior precipitation has a much higher resolution than the raw MERRA2 input; for example, ridge and valley features are noticeable in Fig. 2c whereas the field is very coarse in Fig. 2a, and smooth and unresolved in the interpolated field of Fig. 2b. Furthermore, the raw MERRA2 input fails to capture the expected orographic effect whereas the posterior precipitation clearly shows more precipitation at higher elevations in the north and along the watershed ridgelines. The posterior $b$'s, which are informed by the pixel-wise fSCA ablation time series in the reanalysis (assimilation) step, are the multiplicative (bias correction) factors that bring out these features in the posterior precipitation. They contain both a spatially distributed pattern relating to topography and static physiographic features (Fig. 2d) as well as an interannual variation (Fig. 2e, f). We observe that, for the study domain, this interannual variation is correlated with the prior MERRA2 precipitation level: a higher watershed-average prior precipitation correlates with a lower watershed-average bias correction (R = -0.6) (Fig. 2f).



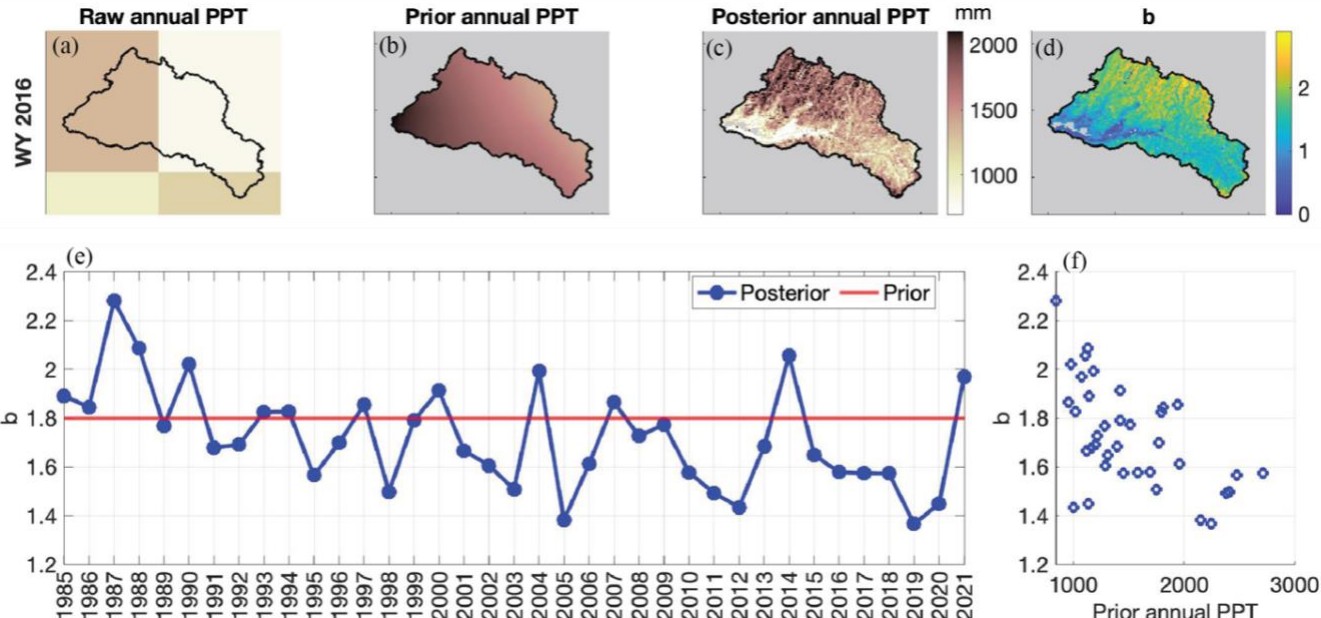

**Figure 2.** For sample WY 2016: (a) Raw MERRA2 annual precipitation at its original resolution. (b) Ensemble-mean annual prior
precipitation, which uses the raw MERRA2 precipitation interpolated to the model resolution. (c) Ensemble-mean annual posterior
precipitation. (d) Ensemble-mean of the posterior bias correction $b$. (e) Watershed-average ensemble-mean prior (red) and posterior
(blue) $b$ per the reference dataset. (f) Scatter plot showing negative correlation between watershed-average annual prior
precipitation and watershed-average posterior $b$. Note that for (c) and (d), a non-seasonal snow mask screens out model pixels located
below 1,500 m, with less than 2 cm of climatological SWE, and/or categorized as glacier.

**2.4 Design of real-time modelling and data assimilation experiments**

We use the $b$ database from the historical reference dataset to inform and develop value-added precipitation bias
correction for the real-time experiments. The list and characteristics of these experiments are tabulated in Table 1. Section
2.4.1 describes the bias correction approach for each experiment. Sections 2.4.2 and 2.4.3 provide further details about the
data assimilation experiments (fSCA and snow depth, respectively). Figure 3 provides an illustrative schematic of the
precipitation bias correction, data assimilation, and resulting SWE estimation for a sample pixel (location A in Fig. 1a) and
water year (WY 2017).

Note here that we are applying and evaluating these experiments in hindcast mode and are selecting 1 April as the
target representative date. As such, we are testing these methods at the end of the accumulation season, when real-time SWE
estimates would provide the most value for water supply forecasts. For true real-time application in an operational context,
other factors such as data latency and computation time should be considered.

Each experiment is evaluated by its ability to reproduce SWE spatial fields as compared to the historical reference
dataset on 1 April, and ASO-derived SWE on the validation day closest to 1 April. We select 1 April because it has traditionally
been used to approximate peak SWE in the Sierra Nevada and is when the key April-July water supply forecasts are made
(e.g., He et al., 2016). For both, we compute three metrics: Pearson correlation coefficient R to quantify how closely the



reference spatial distribution is captured, normalized root mean square difference (normalized by the observational mean, NRMSD, %) to measure bias and random error, and mean difference (MD, mm) to measure the average bias.

Note that the Uncorrected, Uniform, Case A and Case B experiments (Table 1) only use the forward-modelling component of the reanalysis framework; because there is no data assimilation, there is no reanalysis step and therefore no posterior estimates. Instead, validation of these experiments is performed on the modelling-only prior estimates. The Case B

+ fSCA and Case B + SD experiments include data assimilation and thus yield posterior estimates.

**Table 1. Summary of the methods applied to the six real-time experiments. The listed bias correction represents the mean of the prior $b$ distribution. The "(x)" notation refers to the bias correction being a spatially distributed field, where x is each pixel in the watershed.**

| | Experiment name | | Mean bias correction |
|---|---|---|---|
| **Forward modelling experiments** | Uncorrected | | 1 |
| | Uniform | | 1.8 |
| | Historically informed | Case A | $b_{clim}(x)$ |
| | | Case B | $b_{wet}(x)$, $b_{normal}(x)$, $b_{dry}(x)$ |
| **Data assimilation experiments** | Case B + fSCA* | | $b_{wet}(x)$, $b_{normal}(x)$, $b_{dry}(x)$ |
| | Case B + SD* | | $b_{wet}(x)$, $b_{normal}(x)$, $b_{dry}(x)$ |

* In assimilation experiments, only observations up to 1 April is included.

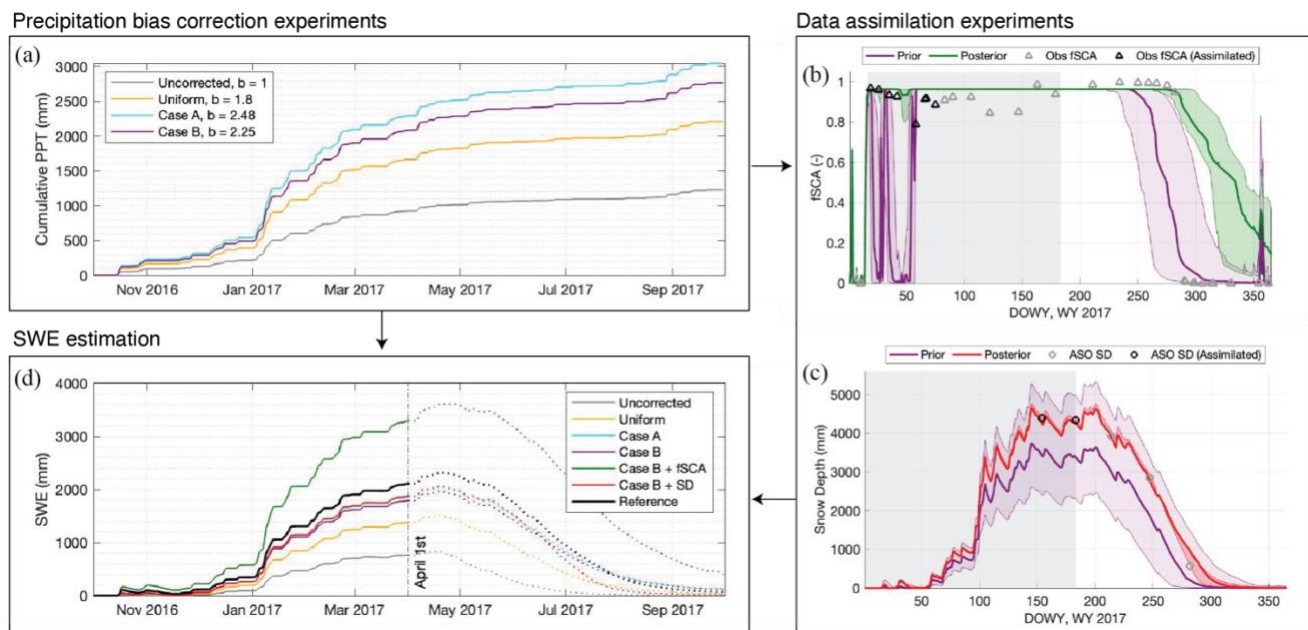


**Figure 3. For a sample wet year (WY 2017) and sample pixel A: (a) Ensemble-mean cumulative prior precipitation for the four forward modelling experiments. (b) Example of how fSCA assimilation is applied. The prior is the Case B experiment, and the posterior the Case B + fSCA experiment. Note that both the median and interquartile range (IQR) for the prior and posterior ensembles are plotted. The assimilation window, indicated with a grey rectangle, ranges from the snow onset date to 1 April.**
**Observations that fall within the assimilation window and on a day when the prior ensemble is non-zero are labelled "assimilated". (c) Example of how snow depth assimilation is applied. Like (b), the prior is the Case B experiment, and the posterior is the Case B + SD experiment. The assimilation window ranges from the start of the water year to 1 April. (e) Ensemble-mean SWE time series for the four forward modelling experiments, two data assimilation experiments, and the historical reference dataset. The outcome of these experiments is evaluated on 1 April.**





### 2.4.1 Definition of precipitation bias correction factors

We generate a baseline, uninformed case where the prior precipitation is uncorrected (Uncorrected in Table 1, Fig. 3a). The Uniform experiment adjusts prior precipitation with a uniform (in time and space) mean prior $b$ that matches that used in the historical reference dataset and defined in Fang et al. (2022) (Table 1). This represents the case for a simple precipitation bias correction (Fig. 3). Note that we maintain the nominal CV and minimum/maximum values from Fang et al., 2022 for the prior $b$ ensemble for this and all subsequent experiments; that is, we alter only the ensemble mean.

The more informed experiments leverage the database of historical $b$ factors generated as a byproduct of the SWE reanalysis framework. They vary from the less-informed cases (Uncorrected, Uniform) in two key ways: the prescribed precipitation bias corrections are spatially distributed, and historically informed. From the historical reference, we compute a spatially distributed climatological $b$ for each water year, withholding the $b$ value from a given year in deriving a long-term climatology for that year. These climatological values are used as the mean bias correction for the Case A experiment (Table 1, Fig. 3a). Because we observe a relationship between precipitation level and watershed-average $b$ in the historical reference (Fig. 2f), we also derive a bias correction that is conditioned on water year type. For each water year, we determine a type based on the historical prior precipitation (cumulative on 1 April), where <30th percentile is "dry", >70th percentile is "wet", and in between is "normal". We take a spatially distributed average of the historical $b$'s of all the other years classified in that water year type; that average becomes the mean value of bias correction for that year in the Case B experiment (Table 1, Fig. 3a). In Fig. 4, we illustrate how the historical $b$ factors across the watershed for wet years tends to be less than the climatological values (with the exception of headwater river valley bottoms, as shown in red in Fig. 4d); and those for dry years tend to be greater (Fig. 4b). In Fig. 3a, we see how the prior precipitation in Case B differs from Case A because of the difference in precipitation bias correction.

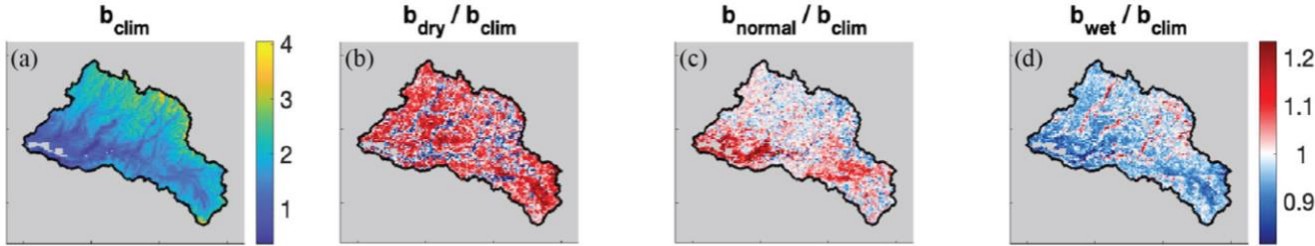

**Figure 4. Per the historical reference: (a) Climatological (WY 1985-2021) bias correction map. (b) The ratio between the dry year and climatological bias correction. (c) Same as (b) but for normal years. (d) Same, but for wet years.**

The last two experiments include the assimilation of either fSCA or snow depth measurements; these experiments are subsequently referred to as Case B + fSCA and Case B + SD, respectively. The data assimilation methods are described further in sections 2.4.2 and 2.4.3. We choose to use the Case B method of correcting prior precipitation for these experiments because it represents the most informed and sophisticated bias correction.





### 2.4.2 Assimilation of fSCA observations

Previous data assimilation experiments that use fSCA to effectively improve SWE estimates have typically utilized measurements from both the accumulation and melt seasons (e.g., Girotto et al., 2014; Margulis et al., 2016; Fang et al., 2022),
which are assimilated retrospectively in a single batch at the end of the water year. This is done with the understanding that it is the fSCA ablation time series combined with estimates of snowmelt that is most directly connected to SWE. The value of fSCA measurements during the accumulation season is expected to be more limited because snow coverage is often complete or near-complete (i.e., fSCA = 1) when snow is accumulating in snow-dominated areas. Past studies have found limited to no improvement from prior modeled SWE estimates when assimilating fSCA observations over the accumulation season,
independently from one another, or at sites which experience long periods of near-complete snow cover (Andreadis and Lettenmaier, 2006; De Lannoy et al., 2010).

Here, we test whether there is any additional benefit in SWE estimation with bias-corrected precipitation when assimilating fSCA up to 1 April, and where in the watershed that benefit might be the greatest. We use the same methods described in section 2.3 to derive, screen, and assimilate fSCA observations, but only include the subset of fSCA observations
that fall between a snow onset date and 1 April (Fig. 3b, more details about the assimilation window in Text S1). Note that not all pixels in the watershed assimilate the same number of fSCA observations for a given year because of differences in the snow onset date, cloud cover, and satellite orbital patterns.

### 2.4.3 Assimilation of snow depth observations

The experiment with snow depth assimilation (Case B + SD) incorporates multitemporal lidar-derived snow depth
observations taken over the Tuolumne watershed by ASO (Painter et al., 2016) on and before 1 April. The observations cover the entire watershed (Fig. S3). In contrast to fSCA, snow depth observations are expected to provide more insight into accumulation season SWE because of the close relationship between snow depth and SWE. Margulis et al. (2019) demonstrated how the assimilation of even a single day of ASO snow depth observations was able to significantly improve posterior estimates of SWE later in the year. Here, we seek to quantify how much assimilating snow depth observations could improve upon SWE
estimates that already incorporate a historically informed precipitation bias correction. We use data for three representative years: WY 2015 (dry), 2016 (near average), and 2017 (wet). The number and dates of the ASO observations used for assimilation and validation purposes are summarized in Table 2. Prior to assimilation, the 50 m ASO snow depth product was regridded to the modelling resolution. Following Margulis et al. (2019), we specify a measurement error standard deviation of 5 cm. Figure 3c illustrates snow depth assimilation for a sample model pixel: the observations on March 3rd and 1 April of this
year fall within the prior ensemble, and so are assimilated and yield a higher posterior mean and narrower ensemble.

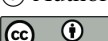



**Table 2. ASO acquisition DOWYs and their corresponding dates. The dates of the observations used for assimilation and for**
**validation (i.e.: the day closest to 1 April) are indicated for each WY.**

| | | Assimilated | Validation |
|---|---|---|---|
| **WY** | **2015** | 140 (Feb 17), 156 (Mar 5), 176 (Mar 25) | 185 (Apr 3) |
| | **2016** | 178 (Mar 26), 184 (Apr 1) | 190 (Apr 7) |
| | **2017** | 154 (Mar 3), 183 (Apr 1) | 214 (May 2) |

**2.5 Connection to streamflow**

We further evaluate the real-time SWE experiments by their ability to yield snowmelt estimates and streamflow forecasts that match observed streamflow at the TGC (Tuolumne River at the Grand Canyon) USGS gauges. Note that for this comparison, the focus is on a subwatershed that drains through the TGC gauge (Fig. 1a). Continuous daily streamflow records
are generated from observations at the USGS gauge TGC for WYs 2009-2021 (Text S2).

We measure how well daily watershed-average snowmelt estimates correlate with daily observed streamflow for the key forecasting period April-July with Pearson correlation coefficient R. For each experiment, the SWE reanalysis framework is run forward in time from 1 April with the known meteorological forcings of that year, yielding "perfect" hindcasts for SWE and snowmelt. Snowmelt is estimated as the negative daily changes in SWE, assuming sublimation is negligible.

We quantify the ability of watershed-average estimated 1 April SWE to predict April-July (AJ) streamflow volume with a historical linear regression. For a given year, we build a regression from all other years (i.e., excluding the one being evaluated) using observed AJ streamflow as the predictand (Fig. S8). Observed AJ streamflow from WY 2019 is excluded because of incomplete observations (Fig. S2). We quantify the bias and mean absolute differences between the predicted volumes from the experiments and the reference. Because we are treating the historical reference dataset as the ground truth
in this study, the predicted volume from its SWE estimates is treated as the "best case" prediction and so is the target in this comparison. The relationship between estimated 1 April SWE and observed streamflow is affected by other factors like rainfall and soil moisture conditions which make errors between the predicted and observed streamflow volume inevitable, and not the focus of the study. Note that for this analysis, the snow depth experiment is excluded because it only has 3 years of results.

**3 Results**

**3.1 Value added from historically informed precipitation bias correction**

Maps of 1 April SWE estimates and their difference relative to the historical reference (Fig. 5) highlight key differences across experiments; in particular, these show how using a historically informed precipitation bias correction (Case A and Case B) yields 1 April SWE that better matches the reference. Here, the reference is the posterior SWE estimates from the historical reference dataset which is constrained by the full set of fSCA observations across the water year. Notably, we
can see how Case A and Case B produce 1 April SWE distributions that are much better spatially-resolved than the less-informed Uncorrected and Uniform experiments: individual ridges and valleys are more prominent and match the historical





reference more closely (Fig. 5a). This is also notable in the maps illustrating the bias in SWE: in the Uniform experiment, a strong geographical pattern exists in WY 2016 and 2017, where the lower-elevation areas of the watershed (generally, the southern half) consistently show a positive bias in SWE estimation, and the higher elevations show a negative bias (Fig. 5b). This geographical distinction is lessened in Case A and Case B in 2017 and 2015, and eliminated in 2016. This illustrates that the historically informed, spatially distributed bias correction is able to correct biases in input precipitation that are related to elevation and topography, whereas the uniform bias correction smooths over these spatial differences. We also see that Case A and Case B yield relatively similar results in terms of the spatial distributions and magnitudes of error.







**Figure 5. (a) Maps of 1 April SWE for the historical reference, Uncorrected baseline, and three experiments for three representative water years: 2015 (dry), 2016 (normal), and 2017 (wet). (b) Maps of the difference in 1 April SWE (experiment - reference) for the same years and experiments. The NRMSD relative to the reference is included. Pixels where both the reference and experiment estimate 0 SWE are greyed out in addition to the mask in these maps.**

All modelling-only experiments (Uniform, Case A, Case B) outperform the Uncorrected baseline in at least two 1 April SWE metrics (Fig. 6e-g). The greatest reduction in error relative to the Uncorrected baseline occurs when a spatially distributed bias correction is used: on average across all years in the record, Case A and Case B reduce NRMSD by 58% and





57%, improve R by 43%, and reduce bias (MD) by 88% and 85%, respectively (Fig. 6e-g). The limited difference in performance between the two cases suggests that the primary value of the historical database of bias correction distribution lies in its description of the (more or less static) *spatial distribution* of precipitation bias, rather than its temporal patterns or uncertainty. A more simple uniform bias correction is also effective at reducing error but to a lesser degree: Uniform reduces average NRMSD by 35% and MD by 97%, and yields an insignificant average impact on R (Fig. 6e-g). Note that the MD metric averages values across the watershed and so does not represent the spatial spread of error in these estimates (Fig. 5b). For the Uniform experiment in particular, a lower watershed-average MD masks high positive and negative errors across the watershed (Fig. 5b).

The performance of these experiments varies by year. For example, in WY 2015, which was historically dry in the Hetch Hetchy watershed, NRMSD is the highest across all experiments (Fig. 6b). Although the two spatially distributed precipitation bias corrections yield similar results in most cases, Case B (where the bias correction is also differentiated by water year type) has an 8-30% lower bias than Case A in low and normal snow years (Fig. 6g). This indicates that differentiating the bias correction by water year type further reduces bias in years with lower snow accumulation. In those years, the bias correction is generally higher than the climatological mean (Fig. 2b), which, when applied to prior precipitation, effectively increases the snowfall input and reduces the negative SWE bias. On the other hand, Case B has a more negative SWE bias in high snow years than Case A (Fig. 6g); in these high snow years, the bias correction is generally lower than the climatological value and so, when applied to prior precipitation, would reduce input snowfall. The impact of a uniform bias correction factor also varies slightly by water year type: Uniform improves R from the Uncorrected baseline in low snow years, but reduces R in normal or high snow years, indicating a poorer spatial correlation to the reference in years with higher snow accumulation (Fig. 6f). High snow years in Uniform are also the only years to show an average positive bias (Fig. 6g), suggesting the uniform bias correction generates higher input precipitation than the reference dataset in higher snow years. This is consistent with the observation in Fig. 2f that, historically, wetter years correlate with a lower watershed-average bias correction; in these years, the posterior bias correction is less than the uniform value.






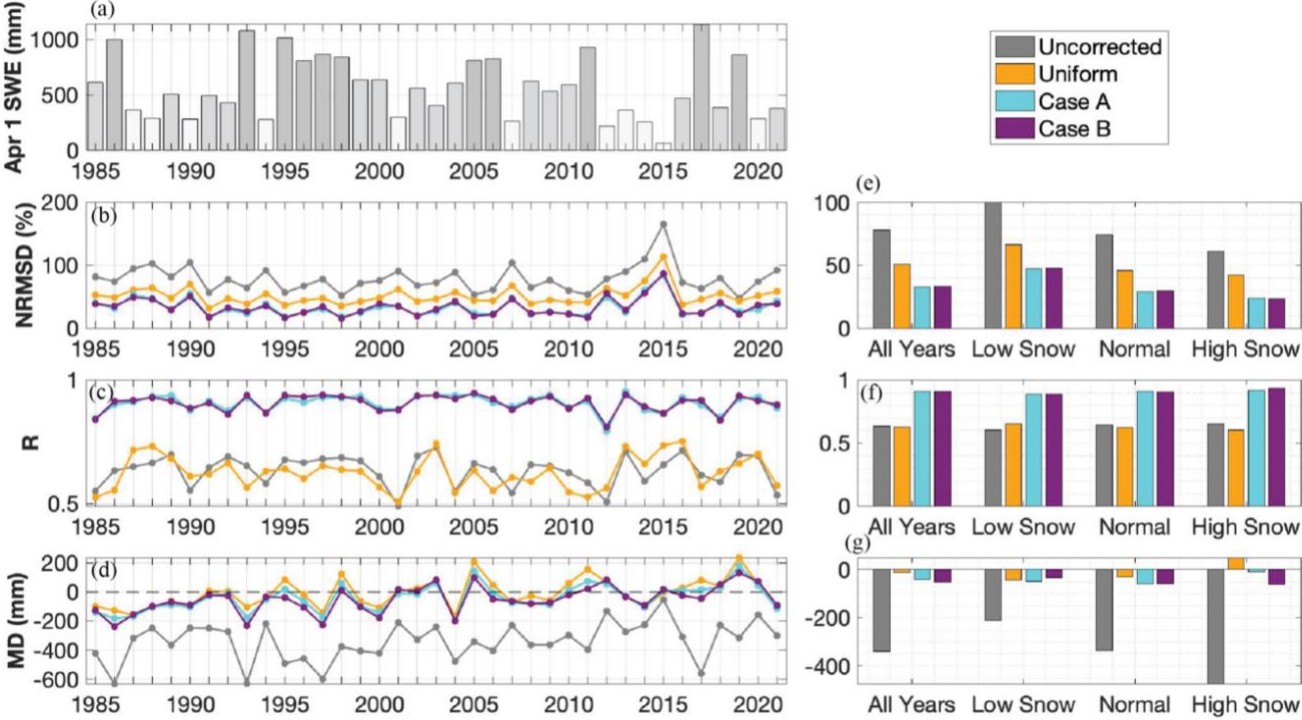

**Figure 6. (a) Watershed-average 1 April SWE per the historical reference dataset. Low snow (below 30th percentile), normal snow, and high snow (above 70th percentile) years are coloured by increasing shades of grey. (b-d) Yearly performance metrics (NRMSD, R, MD; top to bottom) for 1 April SWE in the four modelling-only experiments as compared to the reference. (e-g) Performance metrics (NRMSD, R, MD; top to bottom) averaged across all years, low snow years, normal snow years, and high snow years.**

Figure 7 illustrates the distribution of error in climatological SWE in these experiments across elevation bands. At low elevations (below 2600 m), the Uncorrected baseline tends to have the least error (lowest RMSD), but also a significant negative bias (MD) that persists across all elevations and indicates an underestimation of SWE (Fig. 7b, d). The other three experiments, which incorporate varying levels of precipitation bias correction, instead overestimate SWE at low elevations: Case A and Case B by small magnitudes (~5-10 mm) and Uniform by a greater magnitude (~200 mm) (Fig. 7d). Spatial

distribution of SWE is best represented by Case A (with Case B a close second) at low elevations, as indicated by the highest R (Fig. 7c). At mid elevations (between 2600 m and 3000 m), the lowest error (RMSD) occurs in the Uniform experiment, then Case B and Case A, and then Uncorrected (Fig. 7b). The bias in Case A and Case B is consistently negative at mid elevations, with Case A having the slightly lower magnitudes (Fig. 7d). The bias in the Uniform experiment switches from positive to negative at around 2760 m (Fig. 7d); this trend demonstrates cancellation effects happening in the watershed-

averaged bias, where the watershed-average value in Fig. 6 may be low despite both high positive and negative biases at different elevations. Error across all three performance metrics increases more steadily and more steeply with increasing elevation in the Uncorrected baseline than the Case A or Case B experiments (Fig. 7b-d). This implies that the spatially distributed precipitation bias correction applied to the latter two is effective at reducing error across the watershed, especially





at higher elevations where most of the SWE is located. At high elevations (above 3000 m), the lowest error (RMSD) and bias
(MD) occurs in Case A, with Case B as a close second and Uncorrected as significantly worse (Fig. 7b, d). The spatial
distribution of SWE (R) is best portrayed by Case B, with Case A close behind (Fig. 7c). Notably, the elevation band (3100-
3200 m) with the lowest R across all experiments is also the one with the highest climatological April 1 SWE (Fig. 7a, c). This
suggests that this elevation band could benefit from ablation season and post-ablation fSCA assimilation, which is included in
the historical reference. All experiments underestimate SWE at high elevations (Fig. 7d), indicating that the spatially
distributed precipitation bias corrections are not enough to fully compensate for the negative bias in SWE estimation at high
elevations, although it does reduce that bias.

Overall, including a spatially distributed precipitation bias correction significantly improves SWE spatial estimates
across all elevation bands, as indicated by consistently higher R values in Case A and Case B than both the Uncorrected and
Uniform experiments (Fig. 7c). This bias correction also yields error that is more uniform across elevation bands; crucially, it
reduces error more significantly at higher elevations where more SWE accumulates (Fig. 7b, d). It is worth noting that, to the
extent that reanalysis precipitation products such as the MERRA2 input used in these experiments get informed by existing in
situ precipitation gages, those data are generally at lower elevations. This emphasizes the need to get accurate spatially
distributed bias corrections that adjust uncertain precipitation inputs at higher elevations where most SWE accumulates.

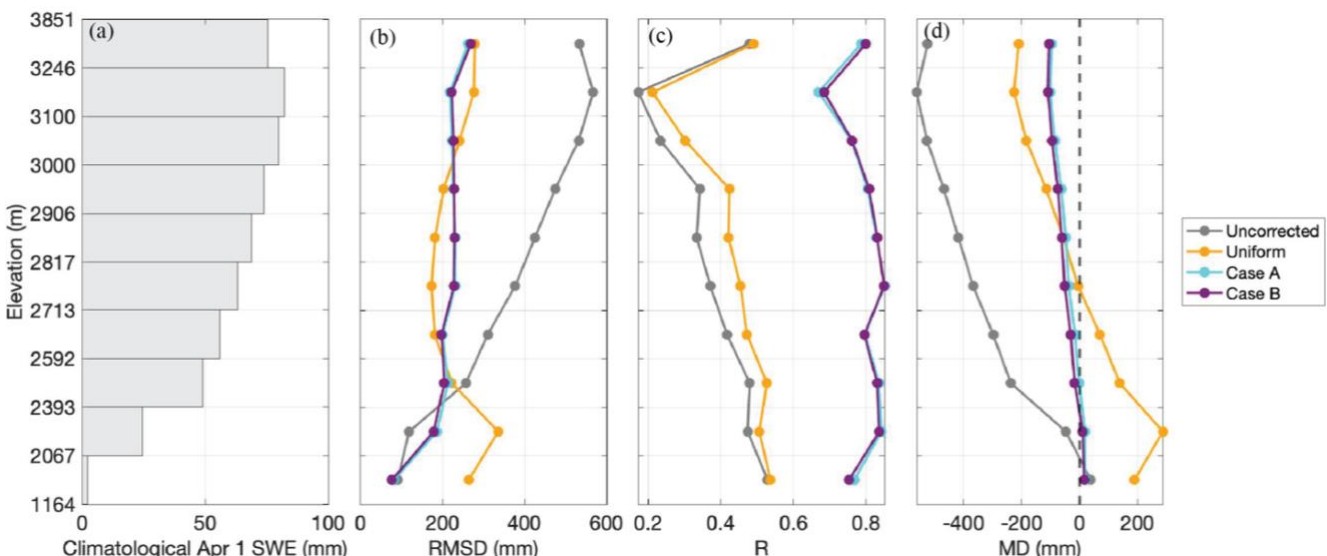

**Figure 7. (a) Long-term (WYs 1985-2021) average 1 April SWE in each of 10 elevation bands of the watershed. Elevation band
bounds were determined by distributing an even number of model pixels into each band. (b-d) Long-term average RMSD, R, and
MD (left to right) for each elevation band for the Uncorrected baseline and three modelling-only experiments.**



### 3.2 Additional value through data assimilation

### 3.2.1 fSCA assimilation

365        The value of assimilating fSCA to estimate SWE lies primarily in its ability to track the loss of snow cover during the melt season, so the expectation for additional insight from assimilating fSCA observations only before 1 April (as was done in the Case B + fSCA experiment) is small. This is confirmed by the results: when looking at the overall impact of fSCA assimilation on 1 April SWE estimates, the watershed-scale NRSMD is reduced from the prior in only 2 of 37 years (Fig. 7). A further 7 years show a posterior NRMSD within only 10% of the prior, indicating limited difference. In the remaining years

(the majority), fSCA assimilation brings the posterior estimates further from the historical reference and increases the NRMSD (Fig. 7). Note that here, the experiment uses the spatially distributed precipitation bias correction from Case B. This implies that accumulation season fSCA observations, which comprise most of the observations before 1 April, are more noisy than helpful in data assimilation. In fact, assimilating these observations is detrimental to overall SWE accuracy in most years (Fig. 7). Overall, this method is not a useful approach to improving real-time 1 April SWE, especially when the precipitation input

is already bias-corrected as is the case here.

We find that the timing of fSCA observations in the water year is significant to determining whether their assimilation reduces error in SWE estimates. For example, in WY 1988, only 1-2 fSCA observations were assimilated into the fSCA experiment, but all of these occurred after pixel-wise peak SWE and before 1 April (Fig. S4a). In that year, 70% of pixels showed a reduction in posterior error relative to the reference, and the watershed-scale NRMSD was reduced 5% (Fig. 7). In

WY 2012, pixel-wise peak SWE averaged after 1 April for the watershed, but the NRMSD was still reduced (by 21%) with fSCA assimilation because enough fSCA observations (6-10) were assimilated at low-elevation pixels (Fig. S4b). Note that in cases with fewer fSCA observations during the accumulation season, there is often degradation in the posterior SWE estimate.

Individual pixels in the watershed can show improvement with fSCA assimilation even when the total error is increased. The absolute bias relative to the reference is reduced at over 50% of the pixels in the watershed with fSCA

assimilation in 7 years (Fig. 7). The pixels where fSCA assimilation reduces error tend to have an earlier peak SWE, lower elevation, and higher number of fSCA observations assimilated (Fig. S5). In most years, there is a statistically significant difference in all these metrics between the pixels with an improvement and those without an improvement. This is consistent with the finding in Andreadis and Lettenmaier (2006) that improvements in SWE estimation from fSCA assimilation are more evident at lower elevations and during snowmelt.





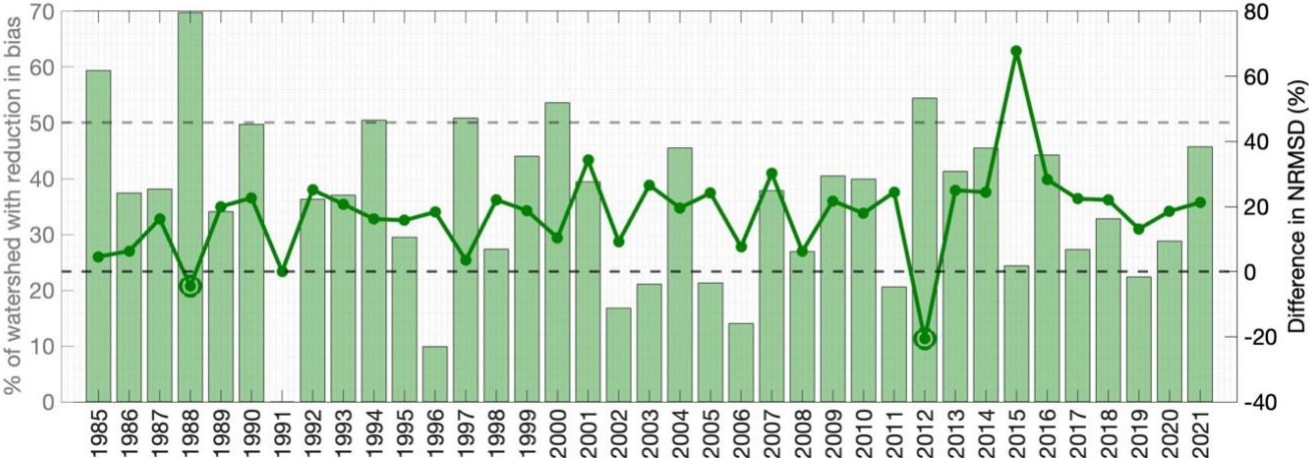


**Figure 8. The height of the bar indicates the percent of pixels that show a reduction in error (absolute difference) relative to the reference when fSCA assimilation is applied for that WY (left axis). The solid line indicates the differences in watershed-scale NRMSD relative to the reference between the prior and posterior estimates (i.e., NRSMD$_{post}$ - NRMSD$_{prior}$) (right axis). The plot**
**points below 0 are highlighted with an extra circle; in these years, the posterior estimate yields a lower NRMSD than the prior. The grey dashed line indicates the point at which 50% of the watershed shows a reduction in bias (left axis); the black dashed line indicates where the posterior NRMSD is less than the prior (right axis). Note that in 1991, no fSCA observations were assimilated so there was no change from the prior to the posterior SWE estimates.**

### 3.2.2 Snow depth assimilation

In addition to mostly reducing watershed-scale average error (as expected), snow depth assimilation brings more
spatial heterogeneity to SWE estimates and reduces pixel-wise bias. Note that here, the reference is SWE from the ASO observations succeeding the last one that was assimilated; that is, day of water year (DOWY) 185 in WY 2015, 190 in 2016, and 214 in 2017. In the maps of Fig. 9, we observe that in WY 2016 and 2017, the posterior estimates show a wider range in SWE than the prior, with higher SWE estimates at higher elevation pixels. Note that this prior-posterior pair uses the historically informed spatially distributed bias correction from Case B. In WY 2015, we observe how most of the watershed
has already lost its snow cover by the day of comparison according to the posterior estimates (Fig. 9a), which contributes to the poor performance of snow depth assimilation in this year.

In both WY 2016 and 2017, we observe a decrease in pixel-wise error with snow depth assimilation. In 2016, most of the watershed demonstrates an underestimation of reference (ASO) SWE in the prior; this is lessened in the posterior, with some valleys showing slight overestimation (Fig. 9k, l). In 2017, the prior shows an underestimation of ASO SWE in the
northern high-elevation region of the watershed and an overestimation in the southeastern headwaters and low river valley area, some of where ASO observes no snow by this time. In the posterior, the magnitude of pixel-wise error is universally decreased, more of the watershed underestimates ASO SWE, and the lack of snow in the low river valley areas is more correctly captured (Fig. 9h, i). In 2015, most of the watershed except for higher elevation ridge areas in the north shows an overestimation of prior SWE, including areas where ASO observes no snow cover (Fig. 9j). The posterior SWE map shows an almost universal
underestimation of observed ASO SWE in this year, except for the areas where the estimates correctly predict no snow cover





(Fig. 9g). This is consistent with the higher watershed-scale errors observed in the estimations with snow depth assimilation on the validation day in 2015 from Fig. 10.

**Figure 9. (a-c) Maps of prior SWE for the Case B + SD experiment on the validation day in three water years. (d-f) Maps of posterior SWE for the Case B + SD experiment. (g-i) Maps of the difference between prior SWE and ASO-derived SWE. (j-l) Maps of the difference between the posterior SWE and ASO-derived SWE. Values of NRMSD relative to the reference are listed. Pixels where both the experiment and the reference have 0 SWE are greyed out in addition to the mask.**



In both WYs 2016 and 2017, the Case B + SD experiment shows the lowest NRMSD (81% less than the Uncorrected baseline) and the highest R (Fig. 10a, b). This also holds true for bias (MD) in WY 2016 (Fig. 10c). In all three years, Case B
+ SD shows a negative bias, indicating that SWE estimates are consistently underestimated (Fig. 10c). Assimilating snow depth observations reduces the prior NRMSD by 43-46% and increases the R by 6-12% in WYs 2016 and 2017 (Fig. 10a). Note that for the Case B + SD experiment, Case B represents its prior (i.e., before assimilation). Fig. 10b demonstrates how, for a sample model pixel, the two ASO snow depth observations assimilated in 2016 can bring the prior estimate up to a posterior that better fits the observations both before and after 1 April.

In WY 2015, the outcome of the Case B + SD experiment is different on its validation day. On the days when ASO data is assimilated, the posterior SWE estimates successfully (and as expected) reduce error metrics from the prior (Fig. S6) but by the validation day on DOWY 185, 42% of the basin (mostly higher-elevation higher-snow areas, Fig. S7) has a worse bias in posterior SWE than in the prior. Although overall error is still reduced from the Uncorrected baseline on this day, NRMSD and MD are greater, and R is less, in the posterior than in the prior (Fig. 10). We hypothesize that anomalies in
precipitation input that occur after assimilation explains the poor performance of snow depth assimilation on this day (Fig. 11a). Of the pixels in the watershed that exhibit higher error in the posterior than the prior estimates, 96% experienced an increase in observed snow depth from DOWY 176 (the last day when observations are assimilated) and DOWY 185 (the validation day) (for example, Fig. 11a). Note that, in most cases, this increase in observed snow depth corresponds to a decrease in observed SWE (Fig. 11b); this implies a decrease in ASO-derived snow density, which is consistent with a fresh snowfall
event. The precipitation forcing around those days does not reflect a snowfall event. This inconsistency between precipitation forcing and observations, and the fact that it happened after the last assimilated observations and that WY 2015 was an exceptionally dry year, explains the negative bias at those pixels and in the watershed by the validation day in 2015.





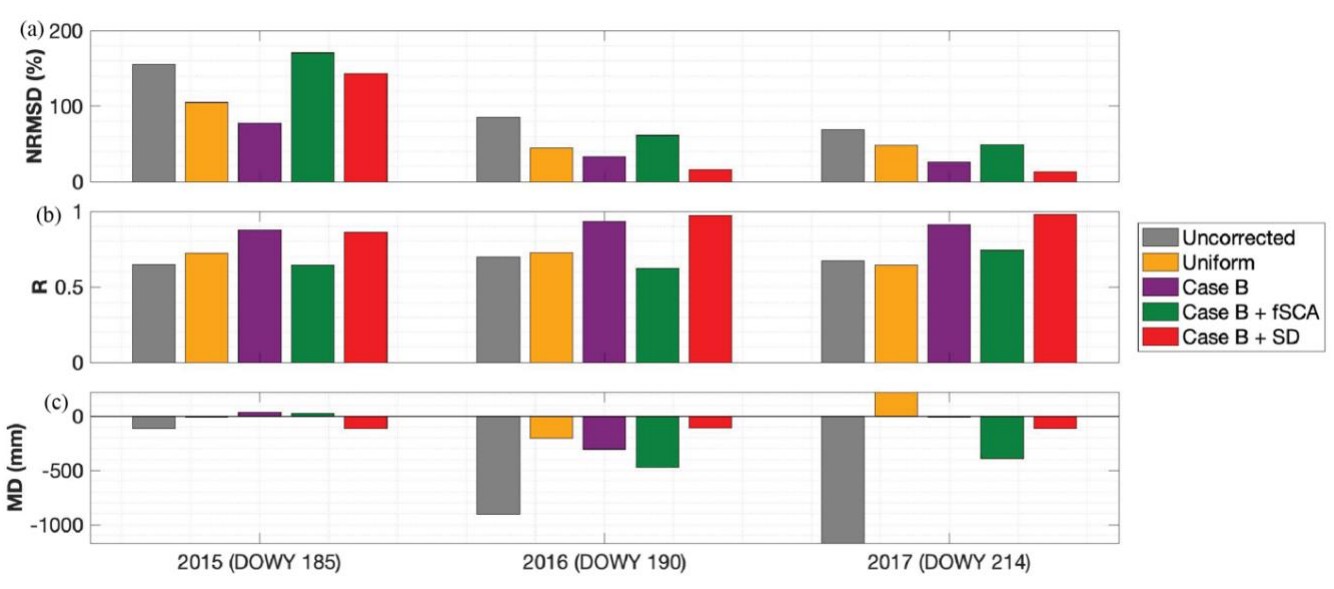

**Figure 10. Summary of performance metrics: (a) NRMSD, (b) R, (c) MD, relative to ASO-derived SWE on the validation day for WY 2015, 2016, and 2017. Because of the similarities between Case A and Case B, only Case B metrics are included here.**

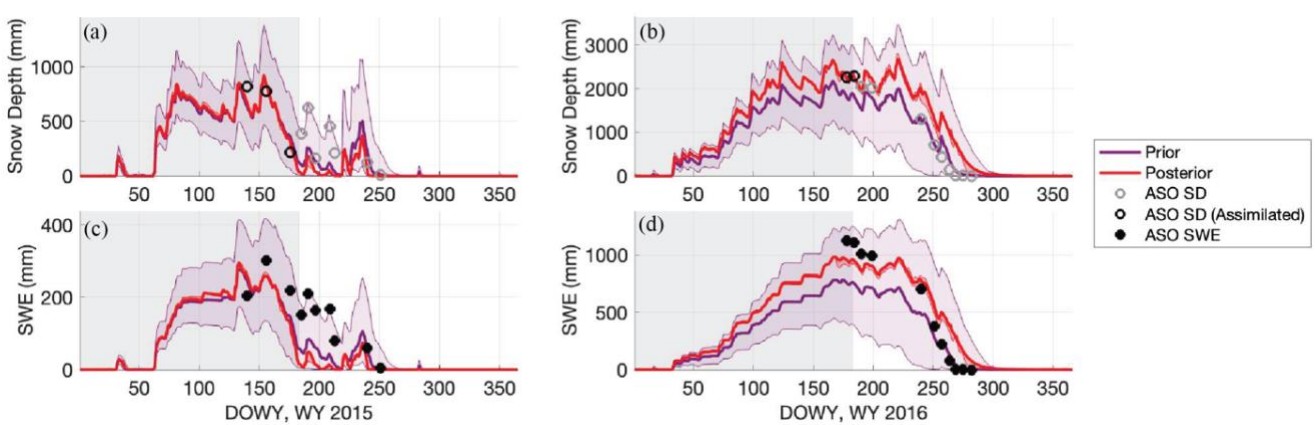

**Figure 11. Sample time series illustrating snow depth assimilation at a sample model pixel in (a) WY 2015 and (b) WY 2016. The prior ensemble is in purple; the posterior in red. Note that both the median and interquartile range (IQR) for the prior and posterior ensembles are plotted. The assimilation window is indicated with a grey background rectangle. The bottom two plots illustrate prior and posterior ensemble mean and IQR for SWE estimates in (c) WY 2015 and (d) WY 2016.**

## 3.3 Value for streamflow forecasting

A key purpose of 1 April SWE estimates is to support streamflow forecasts for spring and summer water supply. We argue that snowmelt is a reasonable proxy for streamflow in this case because Hetch Hetchy is a snow-dominated watershed. We observe that all experiments using bias-corrected precipitation are effective at yielding post-1 April snowmelt estimates





that correlate better to observed streamflow than the Uncorrected baseline. Fig. 12a-f illustrates both the cumulative and daily time series of simulated snowmelt and observed streamflow for April-July in WYs 2015-2017. We observe that the cumulative simulated snowmelt often exceeds the cumulative streamflow (Fig. 12a-c); this is expected because some snowmelt will infiltrate into the soil column or evaporate before reaching the river. Regardless, the Uncorrected baseline consistently yields the least amount of cumulative snowmelt (Fig. 12a-c). In WY 2015, we see that the experiment with snow depth assimilation

also yields cumulative snowmelt that is less than cumulative streamflow (Fig. 12a); this is consistent with the high error and significant underestimation of SWE in that year noted in Fig. 9 and Fig. 10. We observe that the daily simulated snowmelt reasonably captures peaks in observed streamflow: for example, the peak at the end of June 2017 which presumably is driven by a large snowmelt event (Fig. 12f).

How well the simulated daily snowmelt corresponds to observed streamflow is quantified with a correlation

coefficient (R). We test different lag times between snowmelt and streamflow and find the highest correlations with 1 day lag. We find that these correlations are significant and higher than that for the Uncorrected baseline in every year for every bias-corrected experiment (except for Case B + SD in 2015); on average, by 31-39% (Fig. 12g, h). In WY 2016, the highest correlation occurs in the experiment with snow depth assimilation (Fig. 12g). Excluding this experiment because it only yields results for a subset of WYs, the highest average annual correlation coefficient (0.74) is shared amongst Case B and Uniform.

It is worth noting that Uniform has cancellation of errors at the watershed scale; that is, high and low within-watershed biases in 1 April SWE are averaged out (as demonstrated in Fig. 5b). Here, by aggregating snowmelt to watershed-average values, we are likely similarly averaging out within-watershed biases. The experiment with accumulation season fSCA assimilation consistently has lower correlations than those without in all water year types (Fig. 12h). This is consistent with the result that the Case B + fSCA experiment often yields higher errors in 1 April SWE (Fig. 7). In low snow and normal years, the highest

correlation occurs with Case B snowmelt; in high snow and normal years, the highest correlation is with Uniform (Fig. 12h).



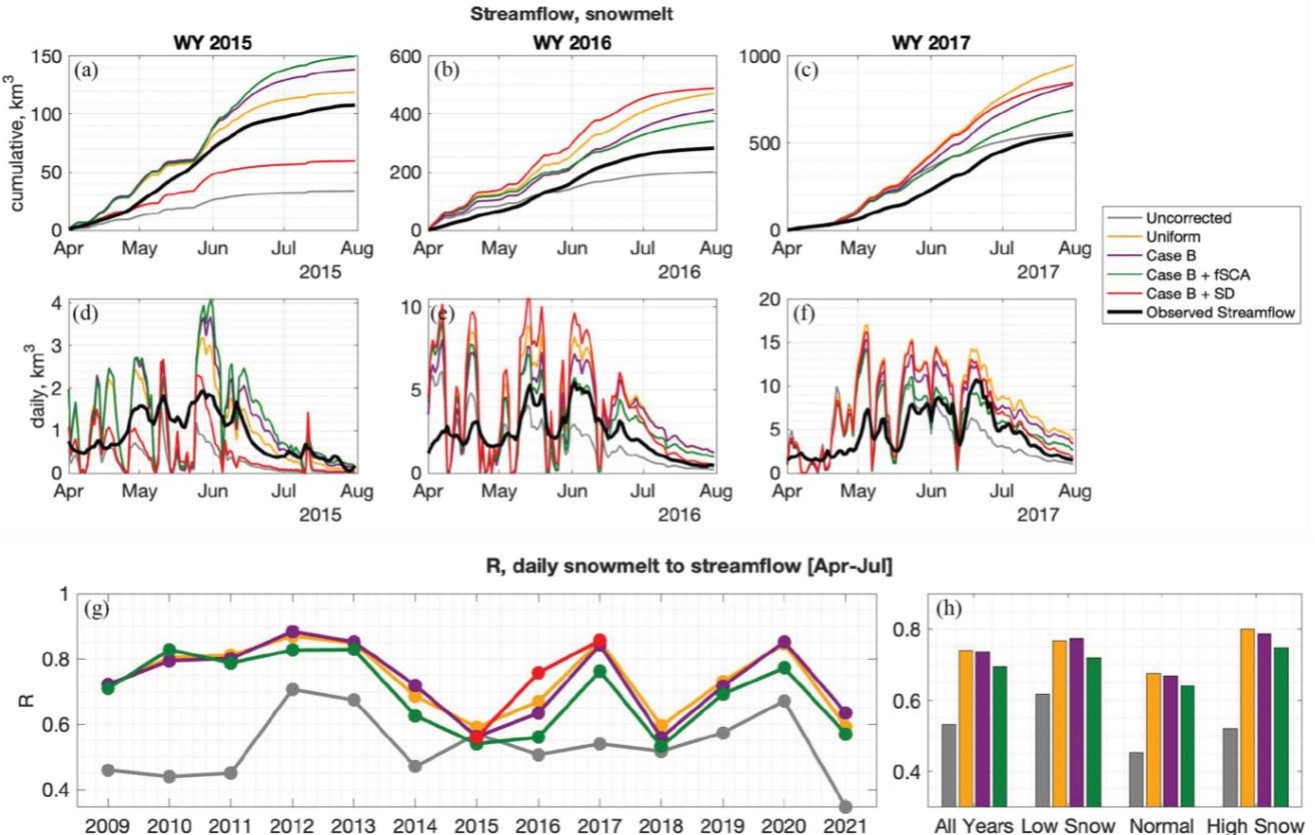

**Figure 12. (a-c). Cumulative daily snowmelt for the Uncorrected baseline and four experiments, and observed streamflow for April-July in WYs 2015, 2016, 2017. Note that because Case A and Case B had similar results, only Case B is included here. (d-f). Same as (a-c) but showing daily values. (g) Correlation R (lag-1) between daily estimated snowmelt and observed streamflow for WYs 2009-2021. (h) Average correlation for all years, low snow years, normal years, and high snow years. Note that these averages exclude the Case B + SD experiment because it only includes 3 years of estimates.**

Figure 13 summarizes results from a linear regression model which predicts April-July (AJ) streamflow volume from watershed-average 1 April estimated SWE. The experiment with snow depth assimilation is excluded because it only has 3 years of results. The multi-year average adjusted $R^2$ for these regression models, developed and applied separately for each experiment-year, range between 0.94 and 0.96 (Fig. S8); these high values emphasize the strong relationship between 1 April SWE and AJ streamflow. In Fig. 13, we compare the predicted streamflow volume from the experiments to the predicted streamflow volume from the historical reference. The predicted AJ streamflow from reference 1 April SWE estimates is considered the "best case" prediction and so is the target in this comparison.

The yearly differences between the experiment and reference AJ streamflow volume range between -30% and 70% (Fig. 13a). The highest differences happen in 2015; this is a historically dry year which also exhibited high errors in 1 April SWE estimates (Fig. 6). Because these yearly differences might cancel each other out in a multi-year average, we also look at the mean absolute differences (MAD) across water year types to gauge the magnitude of error (Fig. 13c). Generally, the bias



and MAD in low snow years is the largest and positive, signifying an overestimation of the reference AJ streamflow volume. The MAD in normal and high snow years are similar in magnitude, but the bias is opposite in direction (positive in high snow and negative in normal) and less in normal years. Overall, the average bias and MAD is reduced from the Uncorrected by the bias-corrected experiments across all water year types (except for Case B + fSCA in high snow years MAD) (Fig. 13a). The greatest improvement in streamflow prediction from the Uncorrected occurs with the Case B 1 April SWE estimates, which reduces average bias by 52% and MAD by 26%. The Uniform experiment is a close second (46% and 25% less average bias and MAD). This is consistent with the error reductions in 1 April SWE observed when using bias-corrected precipitation (Fig. 6). It is worth noting that averaging over the watershed, as is done to obtain the 1 April SWE predictor for AJ streamflow could mask spatially distributed error (which is high in the Uniform experiment, Fig. 5b). A spatially distributed land surface model provides the opportunity to evaluate how improvements in the spatial distribution of SWE, as is observed to happen with spatially distributed historically informed precipitation bias corrections (Fig. 6, Fig. 7), affects runoff modelling and streamflow forecasts.

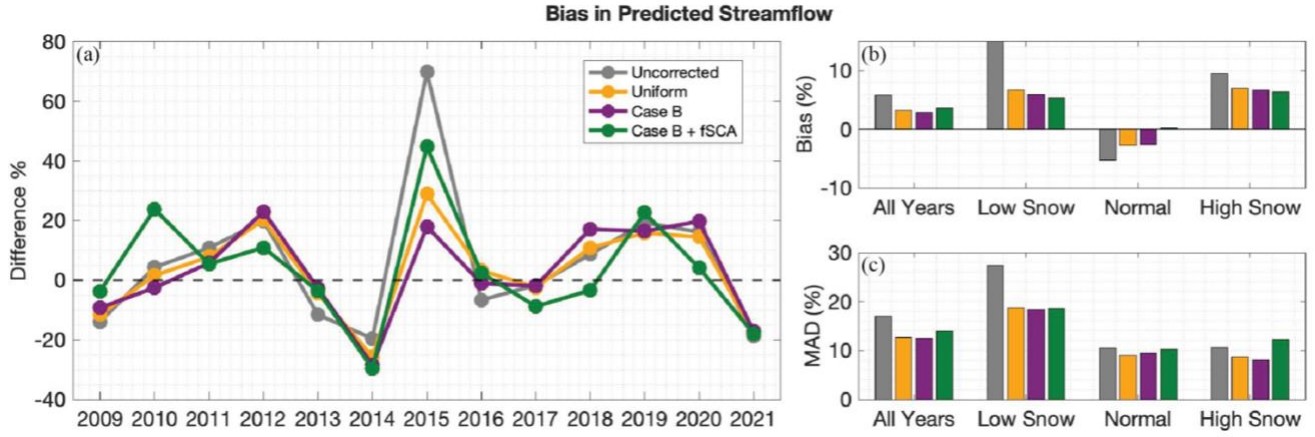

**Figure 13. (a) Difference (%) computed between the predicted April-July streamflow volume of the experiments and the historical reference. (b) Average biases for all years, low snow years, normal years, and high snow years. (c). Mean absolute differences (MAD).**

## 4. Conclusions

Results demonstrate that spatially distributed historically informed precipitation bias correction significantly enhances SWE estimates. With respect to 1 April SWE fields, it reduces error (NRMSD) by 57-58%, increases spatial correlation (R) by 43%, and decreases bias (MD) by 85-88%. A simpler, spatially uniform bias correction (as used as a first guess prior in the original reanalysis methodology) also reduces error relative to uncorrected precipitation but to a lesser degree. We find that the spatially distributed historically informed bias correction yields SWE error that is not only lower but more homogeneous across elevation bands than the uniform bias correction; crucially, it reduces error more significantly at higher elevations where SWE accumulation is greater. It also significantly improves SWE spatial estimates as indicated by higher R



values across all elevation bands. As illustrated by the limited differences in error reduction between the Case A (climatological) and Case B (climatological by water year type) applications, the strength of this approach lies more in its ability to capture the first-order spatial distribution of bias rather than its second-order interannual variability.

Results demonstrate that the assimilation of accumulation season snow depth further improves SWE, whereas fSCA assimilation generally does not. fSCA assimilation prior to 1 April more often degrades than improves posterior SWE estimates, due to the weak relationship between fSCA and SWE outside of the ablation season. In contrast, snow depth assimilation before 1 April leads to a 45% reduction of NRSMD in SWE and a 6% increase in R (excluding the special case of water year 2015). In WY 2015, the precipitation forcing does not capture an observed snowfall event after the last-assimilated snow depth observation; neither the seasonally applied precipitation bias correction nor the assimilation is able to rectify this error. We suggest that assimilation of more observations after single-day anomalies like these could help. This underscores how the assimilation of reliable remote-sensing observations can mitigate forcing anomalies in addition to reducing overall bias and uncertainty. Although remotely-sensed snow depth observations such as those taken by ASO are not readily available everywhere, they are proven to be a good source for improvement in SWE estimation on top of bias correction methods.

The improved SWE estimates provide value for snowmelt-driven streamflow predictions, especially in high snow years. We find that using bias-corrected precipitation reduces average bias in predicted April-July runoff by 46-52% and improves average correlation between daily snowmelt and observed streamflow by 31-39%.

Future work should explore how these methods perform in other global mountain regions. A useful spatially distributed precipitation bias correction in mountain environments can be developed everywhere that historical SWE reanalysis datasets have accurately improved (or are expected to improve) SWE estimates. The power of such an approach lies in the ability to simultaneously downscale and bias-correct globally-available (coarse) precipitation products (e.g. MERRA2 in this work) for use in estimating mountain SWE. Other avenues of investigation could explore more sophisticated methods such as machine learning for bias correction estimation; the assimilation of other sources of real-time snow observations; and the impact of real-time SWE spatial estimates on streamflow forecasts through spatially distributed hydrologic modelling.

**Data availability**

The snow water equivalent (SWE) estimates from the modelling and data assimilation experiments and for the historical reference dataset as well as the historical $b$ bias correction factors are publicly available at 10.5281/zenodo.14014679. Streamflow observations are available from the California Data Exchange Center (CDEC) at https://cdec.water.ca.gov.



**Author contribution**

M.v.K contributed to the development of methodology, funding acquisition, formal analysis, investigation, visualization, and writing. S.M contributed to the conceptualization, funding acquisition, development of methodology, review and editing.

**Competing interests**

The authors declare that they have no conflict of interest.

**Acknowledgments**

The authors would like to acknowledge and thank Dr. Yiwen Fang and Dr. Yufei Liu for their work in preparing the inputs for and applying the reanalysis framework.

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
