# Peer review of "Improved modelling of mountain snowpacks with spatially distributed precipitation bias correction derived from historical reanalysis"

_EGUsphere, 2024_

## Author Comment (AC1)

**To each reviewer:** Thank you for your detailed read of this paper and your insightful feedback. The authors propose to integrate your comments into an improved version of this manuscript. To each comment (which we label and reference with a number), we include our response and any proposed revisions to the manuscript (including text, supplemental information, and figures) in the attached document following this color key. All three reviewer's comments are included in this document; in some cases, we reference another reviewer's numbered comment in our responses.

Green text: proposed new text in the manuscript

Blue text: response to reviewer comment

Black text: original text from manuscript

Gray text: reviewer comments

**Reviewer 1: Michael Matiu**

Von Kaenel and Margulis tackle the notoriously difficult task of estimating spatially resolved SWE in complex mountain terrain by using a by-product of their SWE reanalysis framework to bias correct precipitation in forward SWE modelling. Their approach is an interesting and elegant solution to tackle this issue. However, a large part of their approach is based on the spatial precipitation bias corrections factors in SWE modelling, which is requiring some revisions.

Major points:

1. Conceptually, your bias correction is performing two steps in one: A) downscaling of the coarse-resolution input to a much higher resolution and B) bias correction. This might be explaining some unexpected phenomena like 1) near-zero b factors in valleys, 2) annually varying b factors (assuming intensity-dependent biases of MERRA). I don't expect the authors to change their methodology to test these two steps separately, but maybe the authors find a way to identify these two components. At least, my suggestion is that the authors keep this in mind when discussing results.

Your assessment of the steps involved in the bias correction is correct. In the Case A application, the climatological spatial distribution of the bias correction factors addresses primarily the downscaling step. In the Case B application, we further incorporate inter-annual bias correction by varying the bias correction by water year type, motivated by the relationship we observe between basin-averaged MERRA2 precipitation and the bias correction factor generated through the initial retrospective assimilation of fSCA observations. We propose to edit the following text to section 2.3 (Methods) to clarify this difference:

"They contain both a spatially distributed pattern relating to topography and static physiographic features (Fig. 2d) as well as an interannual variation (Fig. 2e, f). The spatially distributed pattern effectively downscales the coarse-resolution input, and the interannual variation describes intensity-dependent yearly biases of the input (MERRA2 precipitation)."

We also propose adding the following text to section 3.1 (Results):

"This illustrates that the historically informed, spatially distributed bias correction is able to correct biases in input precipitation that are related to elevation and topography, whereas the uniform bias correction smooths over these spatial differences. Effectively, the spatially-distributed bias correction downscales coarse-resolution input."

"The limited difference in performance between the two cases suggests that the primary value of the historical database of bias correction distribution lies in its description of the (more or less static) *spatial distribution* of precipitation bias, rather than its temporal patterns or uncertainty. That is, the historically-informed bias correction is very effective at downscaling coarse-resolution input (as is done in Case A), but less effective at describing interannual bias (as is additionally done in Case B)."

Finally, we propose adding the following text to section 4 (Conclusions):

"As illustrated by the limited differences in error reduction between the Case A (climatological) and Case B (climatological by water year type) applications, the strength of this approach lies more in its ability to capture the first-order spatial distribution of bias rather than its second-order interannual variability. Meaning, the historically-informed and spatially-distributed bias corrections developed herein are an effective tool for downscaling coarse-resolution precipitation input, but less clearly effective at describing its interannual bias."

> 2.  Related: The uniform b fails to capture spatial patterns because your meteo input has effectively only 2 grid cells. I'm not sure how you derived at smooth annual precip like in Fig 2b, but it's spatial patterns are almost inverse to the posterior precip. With a spatially better resolved input (better downscaling?), probably also a uniform b would be better, no? Or maybe not, if the biases depend on elevation or other factors, such as the land surface model?

The smooth annual precipitation in Figure 2b is achieved with interpolation, which is how the prior nominal precipitation is processed in the reanalysis methodology developed in previous work. The reason this is done is that without some level of interpolation, there are stark artifacts that are readily apparent relative to the raw grid and in the prior and posterior. In the reanalysis method, it is assumed in the case of the prior that a downscaling to the finer grid resolution is not known or easily postulated, so the prior is meant to largely be an "uninformed" prior. In the example provided in this figure, the MERRA2 precipitation failed to capture the orographic effect which shows up better in the posterior precipitation, as you observe. This illustrates a case when the bias correction factors are able to correct for an inaccurate spatial distribution of input precipitation, which in this case mostly relates to the low resolution of MERRA2. In other watersheds, biases and errors in MERRA2 may be differently distributed depending on the location, resolution, and broader climatic patterns, and so a uniform b might perform differently. A different data source for the prior precipitation could also be explored (i.e., something based on PRISM for example).

We propose to add the following text to section 2.3 (methods) to clarify the methodology:

"Forcings were sourced from hourly MERRA2 near-surface meteorological forcing data, and the uncertainty models used to perturb input air temperature, precipitation, dew point temperature, and shortwave radiation, as well as model parameters, use the values derived for the Western U.S. domain by Fang et al. (2022) following the methods outlined in Liu and Margulis (2019). The coarse-resolution MERRA2 forcings ($0.5° \times 0.625°$) are first bilinearly interpolated to the model grid resolution (16 arcseconds or ~500m), as is done in the original reanalysis methodology (Fang et al., 2022). Here, we assume that downscaling the prior precipitation to the grid resolution is not known or easily postulated before the reanalysis methodology is applied."

We also propose to add the following text to section 4 (Conclusions):

"The power of such an approach lies in the ability to simultaneously downscale and bias-correct globally-available (coarse) precipitation products (e.g. MERRA2 in this work) for use in estimating mountain SWE. Alternative sources of prior precipitation could also be explored, including products that are already downscaled to a finer resolution (i.e., PRISM)."

3. Discussion is missing.

We are not exactly sure what this comment is referring to; but further points of discussion are added to the Conclusions section (see proposed text in my response to your comment #2, #12, #15).

Minor points:

4. Abstract: Besides relative improvement absolute numbers would also be helpful.

We include absolute values later in the text in figures and propose to add the following to the abstract:

"Results show that spatially distributed historically informed precipitation bias correction significantly improves SWE estimates, reducing the multi-year averaged normalized root mean square difference (NRMSD) from 78% to 33% (-58%) increasing the correlation (R) from 0.63 to 0.9 (+43%) and decreasing mean difference (MD) from -340 mm to -41 mm (-88%)."

We also propose including absolute values (which are shown in Figure 6) in section 3.1:

"The greatest reduction in error relative to the Uncorrected baseline occurs when a spatially distributed bias correction is used: on average across all years in the record, Case A and Case B reduce NRMSD by 58% (to 33%), improve R by 43% (to 0.9), and reduce bias (MD) by 88% (to -41 mm) and 85% (to -52 mm), respectively (Fig. 6e-g).

"The Uniform case reduces average NRMSD by 35% (to 51%) and MD by 97% (to -11 mm), and yields an insignificant average impact on R (Fig. 6e-g)."

5. L35: "Remote sensing…" For mountains in particular, or in general? At least, regarding SWE from remote sensing: mountains no, but in flat terrain, partly yes.

The problem is particularly present in mountainous terrain. We propose adding the following text to this sentence:

"Remote sensing can provide measurements of snow properties like fractional snow-covered area (fSCA, e.g., Selkowitz et al., 2017), snow depth (e.g., Painter et al., 2016), or albedo (e.g., Bair et al., 2019) over large areas, but there is currently no reliable way of measuring SWE from spaceborne platforms, particularly in mountainous terrain (Lettenmaier et al., 2015)."

6. Introduction is missing the state-of-the-art on precipitation bias correction.

We propose to add the following text to the introduction:

"Dynamical and statistical downscaling are two fundamental techniques that translate coarse-scale gridded meteorology to a finer-scale resolution. Statistical downscaling, the less computationally expensive and more common method of the two, relates coarse meteorology fields to high-resolution reference variables and oftentimes inherently includes bias correction (i.e, Gutmann et al., 2014). Recent studies have developed and demonstrated effective machine learning based or statistical downscaling approaches for resolving and/or bias-correcting precipitation fields from satellite-based products (i.e., Sharifi et al., 2019; Lober et al., 2023; Chen et al., 2022; Wang et al., 2023; Zhao 2021; Lu et al, 2020; Ma et al., 2018) and regional climate simulations (i.e, Lie et al., 2021; Velasquez et al., 2020; Yoshikane et al., 2023) in mountainous and moderate-topography regions."

Chen, H., Sun, L., Cifelli, R., and Xie, P.: Deep learning for bias correction of satellite retrievals of orographic precipitation, IEEE Transactions on Geoscience and Remote Sensing, 60, 1–11, https://doi.org/10.1109/TGRS.2021.3105438, 2022.

Gutmann, E., Pruitt, T., Clark, M. P., Brekke, L., Arnold, J. R., Raff, D. A., and Rasmussen, R. M.: An intercomparison of statistical downscaling methods used for water resource assessments in the United States, Water Resources Research, 50, 7167–7186, https://doi.org/10.1002/2014WR015559, 2014.

Li, B., Huang, Y., Du, L., and Wang, D.: Bias correction for precipitation simulated by RegCM4 over the upper reaches of the Yangtze River based on the mixed distribution quantile mapping method, Atmosphere, 12, 1566, https://doi.org/10.3390/atmos12121566, 2021.

Lober, C., Fayne, J., Hashemi, H., and Smith, L. C.: Bias correction of 20 years of IMERG satellite precipitation data over Canada and Alaska, Journal of Hydrology: Regional Studies, 47, 101386, https://doi.org/10.1016/j.ejrh.2023.101386, 2023.

Lu, X. Y., Tang, G. Q., Wang, X. Q., Liu, Y., Wei, M., and Zhang, Y. X.: The development of a two-step merging and downscaling method for satellite precipitation products, Remote Sensing, 12, 398, https://doi.org/10.3390/rs12030398, 2020.

Sharifi, E., Saghafian, B., and Steinacker, R.: Downscaling satellite precipitation estimates with multiple linear regression, artificial neural networks, and spline interpolation techniques, Journal of Geophysical Research: Atmospheres, 124, 789–805, https://doi.org/10.1029/2018JD029241, 2019.

Ma, Z. Q., He, K., Tan, X., Xu, J. T., Fang, W. Z., He, Y., and Hong, Y.: Comparisons of spatially downscaling TMPA and IMERG over the Tibetan Plateau, Remote Sensing, 10, 1883, https://doi.org/10.3390/rs10121883, 2018.

Velasquez, P., Messmer, M., and Raible, C. C.: A new bias-correction method for precipitation over complex terrain suitable for different climate states: A case study using WRF (version

3.8.1), Geoscientific Model Development, 13, 5007–5027, https://doi.org/10.5194/gmd-13-5007-2020, 2020.

Wang, F., Tian, D., and Carroll, M.: Customized deep learning for precipitation bias correction and downscaling, Geoscientific Model Development, 16, 535–556, https://doi.org/10.5194/gmd-16-535-2023, 2023.

Yoshikane, T., and Yoshimura, K.: A downscaling and bias correction method for climate model ensemble simulations of local-scale hourly precipitation, Scientific Reports, 13, 9412, https://doi.org/10.1038/s41598-023-36489-3, 2023.

Zhao, N.: An efficient downscaling scheme for high-resolution precipitation estimates over a high mountainous watershed, Remote Sensing, 13, 234, https://doi.org/10.3390/rs13020234, 2021.

7.  Table 1: unclear what the differences are between b subscripts clim, wet, normal, dry

The subscript "clim" refers to the climatological spatially-distributed b; "wet" refers to the climatological values for only wet years; "normal" for normal years, and "dry" for dry years. The process to derive these is described in 211-216, and we propose adding the following text to the Table 1 caption for further clarification:

"The "(x)" notation refers to the bias correction being a spatially distributed field, where x is each pixel in the watershed; $b_{clim}$ refers to the climatological values over the entire period of record; $b_{wet}$ refers to the average values from only wet years; $b_{normal}$ from only normal years; and $b_{dry}$ for only dry years."

8.  Methodology is missing some information, like what temporal resolution and extent, what spatial resolution, how different spatial resolutions were merged, what regridding methods were used, basics of the land surface model with respect to snow modelling (layers, processes, ...), …

The reference for further description of methodology is Fang et al., 2022. As this study applied essentially the same reanalysis/modeling methodology as that described in this paper, more details were not included in the main manuscript. Section 2.3 outlines how the methodology was applied to develop the historical reference dataset and precipitation bias correction factors. We propose additional text about spatial resolutions and regridding methods in our response to your comment #2. We also propose adding a paragraph to Supplemental Information (SI) to describe more details about the modeling approach with the following text:

"Text S1: Modeling approach

    The reanalysis methodology applied in this study follows the approach described in Fang et al. (2022).  In summary: the land surface model (LSM) used is the spatially-distributed SSiB–SAST model (Sun and Xue, 2001) which simulates a three-layer snowpack when snow depth

exceeds 5 cm, and a one-layer scheme for shallower snow. This model integrates snow depth, snow density, and snowmelt, with mass balance components of rainfall and evaporation. It uses the BATS snow albedo model and the Liston Snow Depletion Curve (SDC) to predict snow-covered areas.

The meteorological inputs include hourly 2-m air temperature, 2-m specific humidity, 10-m zonal and meridional wind speed, surface pressure, surface precipitation, and surface downwelling shortwave from MERRA2. These gridded fields are downscaled from the coarse resolution of 0.5° latitude by 0.625° longitude (~50 km) to the finer model resolution of 15 arcseconds (~500m) using bilinear interpolation and topographic corrections described in Girotto et al. (2014). The topographic corrections include the application of a lapse rate to the temperature field. The input meteorological data are subjected to perturbations to account for biases and uncertainties as follows:

- Precipitation: A lognormal distribution of factors with a mean of 1.80 and a coefficient of variation (CV) of 0.69 (Liu and Margulis, 2019).
- Air Temperature: Normally distributed additive error with a mean of +0.85 K (Girotto et al., 2014).
- Dew Point Temperature: Normally distributed additive error with a mean of −1.37 K (Girotto et al., 2014).
- Shortwave Radiation: Normally distributed multiplicative error with a varying correction factor based on the solar index (SI) (Girotto et al., 2014).

Other land surface model inputs include:

- topographic data from the 30-m Shuttle Radar Topography Mission (SRTM), with gaps filled by the Advanced Spaceborne Thermal Emission and Reflection (ASTER) version 2
- landcover data from the 1-km Advanced Very High Resolution Radiometer (AVHRR) product
- forest cover fraction data from the 30-m Global Land Cover Facility product, specifically the updated Landsat Tree Canopy Version 4.

All inputs are downscaled or aggregated to the model resolution."

9. If I understood correctly, the reanalysis framework is applied in hindcast with assimilation of fSCA to derive b distributions (and the target reference), and then these b are applied in forward models under different experiments. Maybe a flowchart could be helpful in visualizing the overall methodology and when/how different datasets are used at what stage or inherited where.

You understand correctly. We propose to include a simple flowchart to visualize the overall methodology:

[Figure]

10. Fig 2d: b values around 0 in the valleys means that precipitation is effectively removed from those pixels, right? Does this make sense?

A b value of 0 would mean that posterior precipitation is 0 at those pixels. However, the minimum allowed b factor is 0.1 so precipitation is never completely removed. In the historical reference dataset over the study domain, this minimum value is never actually reached and less than 1% of pixel-years have a historical posterior b factor of less than 0.3. So these very low b values are uncommon. Low values of b correspond to significantly reduced snowfall to match the fSCA observations. We propose to add clarifying text about the distribution of b values in section 2.3 as follows:

"Note that for prior precipitation, this uncertainty is quantified by a lognormal distribution of $b$ with nominal mean of 1.8, CV of 0.69, and range of 0.1 to 5, following Fang et al. (2022)."

11. Elevation dependency of biases: I assume you do not bias correct temperature input, only precipitation? Could it be that temperature biases (or their elevation dependency) might impact SWE results?

As part of the reanalysis framework, a constant mean bias correction of +0.85 K (following Fang et al., 2022), and a lapse rate is applied to prior temperature as part of its downscaling to the grid resolution. The temperature input does include an uncertainty characterization, in the form of perturbations (additive factors) applied when creating the prior ensemble. We propose to include these additional details in the new SI text (see response to your comment #8). The bias in temperature inputs will impact SWE accumulation primarily where and when winter temperatures are warmer (i.e., fluctuate at or above freezing level). In this basin, winter temperatures are fairly cold and so the influence of temperature bias will be less than other watersheds with a warmer climate. We do note that the temperature downscaling process, which includes a lapse rate, could introduce spatial bases if the constant lapse rate does not match the true lapse rate. This is a good reason to test this method in other watersheds with different climates. We address a related question about snowfall vs. precipitation correction factors in our response to reviewer 2's comment #2, which includes a proposed text revision.

12. Conclusions read more like a summary. Please think of further implications or generalizations. For instance, mountain precipitation is a challenge everywhere, and your method is rather elegant and does not need in-situ data…

We propose to elaborate on the implications and potential future work related to this study (last paragraph of the conclusions section) with the following text:

"There is the potential for this type of precipitation bias correction in mountain environments everywhere that historical SWE reanalysis datasets have or could be developed, without having to rely on in situ data. The implications of this are widespread. This precipitation bias correction method can lead to improved accuracy of hydrologic models in both research and operational contexts. Because precipitation is a primary driver of hydrologic models, the impact of higher-resolution, higher-accuracy gridded precipitation input is considerable. The value is particularly high in mountainous regions or at elevations with high precipitation uncertainty and/or limited in situ data, which makes other more traditional downscaling methods less applicable."

We propose further additions to the Conclusions section in our responses to your comment #2, #15.

13. Regarding the derivation of precipitation correction factors: What would the differences be if you used observational data (in-situ, or gridded like daymet or similar) to bias correct precipitation like it is done eg in a climate change context?

Observational data is often used for bias correction. The accuracy of that type of bias correction is highest at the point of observation and so is unevenly spread across the watershed. The effectiveness of the precipitation bias correction presented in this study should be more evenly distributed across the watershed; the impact is particularly great at higher elevations where in situ data is less available but snow accumulation is often higher. The challenges with using in situ observations for bias correction (that are not present with the method presented in this study) are that 1) in situ observations, especially of precipitation in mountainous environments, are not always representative of the entire spatial distribution; and 2) many regions do not have access to reliable observations.

14. From an application point of view in streamflow forecasting: The author's approach is conceptually similar to hydrological modelling, where often parameters for precipitation correction are calibrated to match observed streamflows. Maybe the authors could briefly discuss on this.

Parameters for precipitation correction in hydrologic modeling are oftentimes applied uniformly across the modeling domain; that is, without a spatial distribution. But, as you mention, this is a method conceptually similar to our method which corrects precipitation based on satellite observations. We propose a text revision in our response to your comment #12 that addresses the implication of this method for hydrological modeling and streamflow forecasting.

15. Just out of curiosity: Have you tested seasonally varying b factors?

Note that in the original application of the reanalysis methodology, the b factors are derived from the seasonal time series of fSCA observations. In the context of assimilating fSCA, deriving seasonally varying b factors would likely be underdetermined because there is essentially no

instantaneous relationship between b (via SWE) and fSCA. It is the set of fSCA observations during ablation that maps back to SWE (through predicted snowmelt). So it is hypothesized that estimating e.g. monthly b values from a handful of fSCA will not yield a robust seasonal relationship.

It may be possible in the context of the real-time forward modeling to update the b factors as the season progresses and more information is gained about the precipitation/snowfall conditions or other seasonally-varying factors, toward a true seasonally-constant bias correction factor.

We propose to add the following text to the Conclusions to expand on this as a future pathway:

"Other avenues of investigation could explore more sophisticated methods such as machine learning for bias correction estimation; the assimilation of other sources of real-time snow observations; and the impact of real-time SWE spatial estimates on streamflow forecasts through spatially distributed hydrologic modelling. Of particular relevance to the operational context, another pathway for future work is to explore how the derived bias correction, informed by historical relationships between the $b$ factor and precipitation, snowfall conditions, or other factors, could be updated through the season as more real-time information becomes available."

**Reviewer 2: Anonymous Referee #2**

The authors present a data assimilation framework to derive precipitation correction factors, providing improved predictions of snow water equivalent (SWE) at the peak of winter. This research is highly relevant, as such SWE estimates are critical for the reliable management of water resources in catchments reliant on seasonal snowmelt. The methods presented demonstrate promising results, and the study is well-written and provides valuable insights. However, the following questions should be addressed to strengthen the manuscript:

1. Accuracy of the "historical reference" dataset: How accurate are the SWE values in the historical reference dataset? Many of the results rely on comparisons with this dataset, yet an evaluation of its associated errors appears to be missing. Including such an assessment or references to past studies would enhance the robustness of the findings.

The accuracy of the historical reference dataset was evaluated in Fang et al. (2022). We propose to add the following text to section 2.3 to clarify.

"The performance of this historical SWE dataset is evaluated against independent observations in Fang et al. (2022); this verification shows high correlation (0.81 to 0.91) with ASO SWE in the Tuolumne basin (which encompasses the study domain); and good correspondence with in situ SWE in California (root mean square difference = 0.3m, mean difference = -0.15m, correlation = 0.82)."

2. Precipitation correction factors vs. snowfall correction factors: Why did the authors choose to derive precipitation correction factors using snow cover fraction observations instead of deriving snowfall correction factors? Since snow cover fraction observations primarily reflect snowfall events rather than total precipitation, further explanation of this choice is warranted.

You're correct in describing the correction factors as primarily describing snowfall since it uses snow cover fraction observations. The test domain has cold winter temperatures and is at high elevations, and so most precipitation during the winter snow accumulation season falls as snow. This is partly why we're focused on "snow-dominated" watersheds, where rainfall doesn't play a huge role in the annual water budget. To apply more generally, we could separate the two, but that is not something done here. It could be something examined as future work.

We propose to include the following text explaining assumptions involved with this in section 2.2:

"The assimilation of fSCA is adjusting SWE, which mostly depends on snowfall. So while we are multiplying the total precipitation by $b$, we expect that any adjustments are primarily on snowfall. This implicitly assumes that some of the factors leading to snowfall bias, such as

orography, impact rain or snow similarly. This assumption is consistent with previous applications of this reanalysis methodology (i.e., Fang et al., 2022; Liu and Margulis, 2019)."

3. Discrepancy in cumulative observed streamflow: How can the cumulative observed streamflow during the snowmelt period be significantly lower than the model predictions, given that (a) simulated SWE seems underestimated compared to the historical reference dataset, and (b) rainfall is excluded from the modelling? Please provide additional information on evaporation rates (e.g. from the land surface model) to clarify whether these losses could justify the observed differences between measured and modeled cumulative runoff.

We suggest that some of the predicted snowmelt will be lost to evaporation or infiltration, or storage in the form of high alpine lakes in the watershed. Currently we are unable to access evaporation or infiltration rates from our modeling framework; this would instead merit a hydrologic model and we mention this in the Conclusions as a future pathway of work. We acknowledge the differences in magnitude of cumulative predicted snowmelt and observed streamflow; to compare these two data, we look at correlation as a performance metric which should be impervious to such discrepancies. We propose to expand on the existing text in this way to acknowledge this difference:

"We observe that the cumulative simulated snowmelt often exceeds the cumulative streamflow (Fig. 12a-c); this is expected because some snowmelt will infiltrate into the soil column, store in high alpine lakes, evaporate before reaching the river during the April-July time period. We acknowledge this discrepancy by evaluating the correspondence of snowmelt to streamflow with a simple correlation metric; for a more thorough comparison, a hydrologic model is recommended."

We also suggest revising Figure 12 to remove the cumulative streamflow vs. snowmelt plot, which may contain systematic inconsistencies between the measured streamflow and computed snowmelt from SWE. We propose that a hydrological model should be used for that comparison to be fair and complete. Here, we instead focus on the correlation in daily values, which is insensitive to biases due to potential inconsistencies.

[Figure]

Addressing these questions in the manuscript would enhance its clarity and impact. Further detailed comments are outlined below. In summary, this is an excellent study that would benefit from some additional discussion and information to the reader.

**Specific comments**

4. L 44-45: Please add citations from a wider range of researches than those involved in this study to show that data assimilation is a popular method for improving snow modelling results.

We propose to add the following to the Introduction:

"Data assimilation has gained popularity as a way to constrain or correct uncertain model estimates of snow with observations of variables such as fSCA, in situ SWE, or snow depth, and has demonstrated ability as a method to quantify SWE over both melt and accumulation seasons (Magnusson et al., 2014; Margulis et al., 2016; Cortes et al., 2016; Largeron et al., 2020; Liu et al., 2021; Fang et al., 2022; Alonso-González et al., 2022; Aalstad et al., 2018)"

Aalstad, K., Westermann, S., Schuler, T. V., Boike, J., and Bertino, L.: Ensemble-based assimilation of fractional snow-covered area satellite retrievals to estimate the snow distribution at Arctic sites, The Cryosphere, 12, 247–270, https://doi.org/10.5194/tc-12-247-2018, 2018.

Alonso-González, E., Aalstad, K., Baba, M. W., Revuelto, J., López-Moreno, J. I., Fiddes, J., Essery, R., and Gascoin, S.: The Multiple Snow Data Assimilation System (MuSA v1.0), Geoscientific Model Development, 15, 9127–9155, https://doi.org/10.5194/gmd-15-9127-2022, 2022.

Largeron, C., Dumont, M., Morin, S., Boone, A., Lafaysse, M., Metref, S., Cosme, E., Jonas, T., Winstral, A., and Margulis, S. A.: Toward snow cover estimation in mountainous areas using modern data assimilation methods: A review, Frontiers in Earth Science, 8, 325, https://doi.org/10.3389/feart.2020.00325, 2020.

Magnusson, J., Gustafsson, D., Hüsler, F., and Jonas, T.: Assimilation of point SWE data into a distributed snow cover model comparing two contrasting methods, Water Resources Research, 50, 7816–7835, https://doi.org/10.1002/2014WR015302, 2014.

5. Section 1: The introduction lacks a clear outline of the research-gap that warrants another study using a reanalysis framework for improving SWE estimates. What aspects have been covered by existing studies, and what questions remain open? How does the proposed study contribute to existing literature on the topic? Answers to these questions in the text might help readers to better understand the motivation for this study.

This work revolves not around applying another reanalysis framework, but learning from one under the simple premise that a historical observationally-constrained reanalysis should provide

some useful insight for real-time applications. One such example, i.e. precipitation bias correction implicitly derived from a historical reanalysis, is the focus application in this study. We propose to include the following text addition to section 1 (Introduction) to clarify the research gap:

"Getting accurate precipitation fields remains a challenge and a need for modeling snow in mountainous regions. Reanalysis methods that incorporate satellite-based observations have demonstrated effectiveness in estimating SWE in areas where in situ data is sparse. In this paper we leverage a SWE reanalysis framework and historical dataset to address this gap and gain insight into how to improve modeling. Specifically, we derive mountain precipitation bias correction estimates, and develop and test spatially continuous SWE estimates on 1 April."

Reviewer 1 mentioned the need for additional references on state-of-the art precipitation bias corrections which we also propose to include in this context. Our response to reviewer 1's comment #6 includes additional proposed text related to this.

6. Section 2.1: A short summary of snow and meteorological characteristics would be informative. Snow cover duration, annual precipitation and snowfall, average summer/winter temperature and similar variables.

We propose to include the following text addition to section 2.1 (Study domain):

"The study domain comprises the Hetch Hetchy watershed, a headwater catchment for the Tuolumne River in the California Sierra Nevada (Fig. 1). Its drainage area (~1,200 km$^2$) is characterized by complex topography with elevations ranging from 1,150 m to 3,850 m. The watershed is part of Yosemite National Park, which according to the National Park Service receives 95% of its precipitation between October and May, and 75% of that between November and March. During these winter months, temperatures at high alpine watersheds like the study domain typically average below freezing, and snow covers the ground. Snowmelt occurs in the spring and generates water supply that is stored in the downstream reservoir."

7. L 101-102: I think more details about the model are needed, in particular features that are relevant for this study. At what resolution was the simulations performed (this is given later but should be placed here I think)? How was precipitation separated into rain- and snowfall? Is this a single- or multi-layer energy-balance snow model? Currently, the information about the modelling is too sparse.

Reviewer 1's comment #8 also asks for more details about the methodology. In our response to that comment, we include proposed text for an additional SI section that includes more of these details. The reference for more information on the modeling methodology is Fang et al. (2022).

8. L 105: What is the spatial and temporal resolution of MERRA2?

The input MERRA2 forcings are around ~50-60 km spatial resolution and hourly. This can be included as a relevant detail in section 2.2. We address this and propose a text revision in our response to reviewer 1's comment #2.

9. L 112-113: What parameters (mean and standard deviation) were used for this distribution?

Mean of 1.8 and CV of 0.69; this is referenced in Fang et al. (2022). We propose a text revision that includes this detail in our response to reviewer 1's comment #8 and #10.

10. L 123-128: Information about the actual ensemble size and initial conditions needs to be given here. Additionally, it would be helpful to add some brief information about how the meteorological forcing data was interpolated to the model grid.

We propose to include the following text revision to clarify the ensemble size and initial conditions:

"To generate the historical reference SWE dataset for this study, the SWE reanalysis framework was applied to the study domain in the same way as in Fang et al., 2022, but with an increased ensemble size (100) and initial conditions set to default values (for SWE, this is 0 at the start of the water year) to focus on the derivation of posterior $b$ values for testing herein."

We address the question about interpolation in our response to reviewer 1's comment #8.

11. L 177-179: Perhaps move this sentence to the previous paragraph, maybe L 174.

We propose the following text revision:

"Note here that we are applying and evaluating these experiments in hindcast mode and are selecting 1 April as the target representative date. As such, we are testing these methods at the end of the accumulation season, when real-time SWE estimates would provide the most value for water supply forecasts. We select 1 April because it has traditionally been used to approximate peak SWE in the Sierra Nevada and is when the key April-July water supply forecasts are made (e.g., He et al., 2016). For true real-time application in an operational context, other factors such as data latency and computation time should be considered."

12. Section 2.4: I am wondering if I understand the workflow correctly. Step 1: the reanalysis framework (presented in section 2.3) is used to estimate precipitation correction factors by assimilation of all available fSCA observations; an analysis performed for each water year separately. This step leads to precipitation correction factors as displayed in Figure 2. Step 2: in the forward modelling experiments, this "database" of precipitation correction factors is used in the "historically informed" cases, whereas the uncorrected and uniform cases use assumptions about precipitation not derived in section 2.3. Step 3: in the "data assimilation experiments", observations of fSCA and SD are assimilated until 1st of April, on top of the precipitation correction factors. However, if my outline above is correct, the sentence on line 182-184 confuses me, since in the "reanalysis framework", precipitation correction factors are derived by assimilation of fSCA, that are used in Step 2. Thus, the "historically informed" cases actually uses information obtained by data assimilation. Correct or not? If correct, please rephrase the sentence.

Your understanding of the workflow is correct. Reviewer 1 suggested a flowchart diagram to clarify the workflow; we propose this figure in our response to their comment #9. We propose to clarify lines 182-184 to describe that the forward-modelling experiments do not include any "real-time" data assimilation. As you describe, they do depend indirectly on data assimilation through the generation of the precipitation bias correction factors.

"Note that the Uncorrected, Uniform, Case A and Case B experiments (Table 1) only use the forward-modelling component of the reanalysis framework; because there is no direct data assimilation in these experiments, there is no reanalysis step and therefore no posterior estimates."

13. L 203: Does Table 1 refer to the table in the current paper, or that of Fang et al. (2022)?

In the current paper. This can be clarified by reordering the parenthesis:

"The Uniform experiment adjusts prior precipitation with a uniform (in time and space) mean prior $b$ (Table 1) that matches that used in the historical reference dataset and defined in Fang et al. (2022)."

14. L 216-217: Does this mean that the application of the bias correction factors for wet, normal and dry years dampen annual variations in precipitation contained in the original forcing dataset? If so, maybe add a sentence describing this behaviour of the Case B precipitation corrections.

I would not say it dampens the annual variation; the annual variation is still there based on the prior precipitation. Figure 4 describes the behavior of the Case B precipitation corrections (dry vs. normal vs. wet years). Further clarification can be added:

"We take a spatially distributed average of the historical $b$'s of all the other years classified in that water year type; that average becomes the mean value of bias correction for that year in the Case B experiment (Table 1, Fig. 3a). In this way, the bias correction applied in Case B differs by water year type, whereas in Case A it does not."

15. L 231: Please change from "to SWE" to "to maximum/peak SWE" or similar.

We propose the following edit:

"This is done with the understanding that it is the fSCA ablation time series combined with estimates of snowmelt that is most directly connected to peak SWE."

16. L 240-242: Just curious: What is the average number of fSCA observations available per pixel for each year? It seems Figure S4 (not mentioned in the manuscript) and S5 contains this information. Maybe reference that information somewhere in the paper.

The average is about 7 usable observations; this varies based on the year and the Landsat satellites in orbit, as well as factors like cloud cover. We propose to include the following text revision:

"Note that not all pixels in the watershed assimilate the same number of fSCA observations for a given year because of differences in the snow onset date, cloud cover, and satellite orbital patterns. The average in the study domain is about 7 usable observations per year but can vary between 0 to 24."

17. Section 2.5: Why was not snowmelt, liquid precipitation and evaporation from the land surface model used directly here for comparison against observed runoff? How large is the amount of liquid precipitation and evaporation/sublimation from April to August?

We address this and propose text revisions in our response to your comment #3. A hydrologic model would be recommended to provide a more thorough evaluation of the streamflow generation.

18. L 290: Is the "geographical distinction" eliminated in 2016? I still see an overestimation at high altitude and an underestimation at low elevations when looking at Figure 5b.

We propose to reword this sentence to correct:

"This geographical distinction is lessened in Case A and Case B in 2015 and 2016, and greatly reduced in 2017."

19. L 290-294: What is the error of the "historical reference" SWE dataset? The evaluation is performed against this reference. Is the statement "that the historically informed, spatially distributed bias correction is able to correct biases in input precipitation" justified considering that the "historical reference" may contain errors itself? If so, rephrase the sentence, or add a validation of the "historical reference". I assume that the errors of the reference SWE dataset has been assessed in other publications. If so, it would be useful to mention the errors associated with the "historical reference" in section 2.3 and cite relevant literature.

This validation is provided in Fang et al. (2022); we provide a proposed text revision in our response to your comment #1.

20. Section 3.2.2 and caption to Figure 9: I think it would be helpful to not use the word "reference" here to avoid confusion with the "historical reference" used everywhere else. Instead of "reference", please use "ASO-derived SWE" or alike.

This is a good idea and this edit can be made to clarify what the validation dataset is in section 3.2.2. We also acknowledge that there are systematic differences in the ASO snow density estimates and the snow density estimated by our reanalysis methodology. While ASO snow depth observations are reasonably considered the "truth", the need for snow density modeling to estimate SWE from those observations makes ASO SWE prone to error.

"Values of NRMSD relative to the ASO-derived SWE are listed. Pixels where both the experiment and the ASO-derived SWE have 0 SWE are greyed out in addition to the mask."

> 21. L 426: Maybe "posterior" instead of "prior" since snow depth data seem to have been assimilated.

We propose to implement this edit:

"Assimilating snow depth observations reduces the posterior NRMSD by 43-46% and increases the R by 6-12% in WYs 2016 and 2017 (Fig. 10a)."

> 22. L 427-429: What is the meaning of this sentence here? It seems disconnected to the discussion about summary statistics.

This sentence is describing Figure 10b which exemplifies how this improvement looks like for a sample year. This can be moved earlier in the paragraph before the summary statistics.

"Fig. 10b demonstrates how, for a sample model pixel, the two ASO snow depth observations assimilated in 2016 can bring the prior estimate up to a posterior that better fits the observations both before and after 1 April."

> 23. L 457-458: How much water is expected to infiltrate into the soil column or evaporate within this basin? I am particularly surprised by the poor performance in predicting cumulative runoff, as illustrated in Fig. 12a-c. In both 2016 and 2017, most experiments significantly overpredict cumulative runoff. Simultaneously, Figs. 7 and 8 reveal an underestimation of SWE by the Case A and Case B experiments. Furthermore, verification against ASO data, as shown in Fig. 10c, indicates a general underestimation of SWE, with the degree of underestimation varying across experiments. In light of these underestimations, could it be reasonable to attribute these runoff overestimations to water losses due to evaporation and soil infiltration? In particular when considering that rainfall seems to be neglected in these water balance calculations.

We address this in our response to your comment #3.

> 24. L 490: Change "happen" to "occur".

We propose to include the edit:

"The highest differences occur in 2015; this is a historically dry year which also exhibited high errors in 1 April SWE estimates (Fig. 6)."

> 25. L 482-504: What are the regression coefficients? Do they align (e.g., slope less than one) with the results presented in the first part of section 3.3?

The regression models are illustrated in Figure S8. Slopes are less than 1. We propose the following edit to clarify:

"The multi-year average adjusted $R^2$ for these regression models, developed and applied separately for each experiment-year, range between 0.94 and 0.96 (illustrated in Fig. S8); these high values emphasize the strong relationship between 1 April SWE and AJ streamflow."

26. L 530-532: Does this refer to the results presented in Fig. 12 or 13? And what are these improvements compared against?

The first number refers to Figure 13 and the second number refers to Figure 12. The improvements are compared against the Uncorrected baseline. We propose the following revision:

"We find that using bias-corrected precipitation reduces average bias relative to the Uncorrected baseline in predicted April-July runoff by 46-52% and improves average correlation between daily snowmelt and observed streamflow by 31-39%."

**Technical comments**

We propose to integrate these technical edits in the next draft of the manuscript.

L 31: Remove "snow water equivalent".

Figure 3, caption: Should (e) be (d)?

Table 2: What does DOWY mean?

Day of water year. Will be added.

L 347: Picky, but change from 3200 to 3246.

L 368: Change from Fig. 7 to Fig. 8.

L 409: I assume this should be "Fig. 9 h, k" instead of "Fig. 9 k, l".

L 413: I assume this should be "Fig. 9 i, l" instead of "Fig. 9 h, i".

L 414: I assume this should be "Fig. 9 g" instead of "Fig. 9 j".

L 419: I assume this should be "Fig. 9 j" instead of "Fig. 9 g".

L 427: I assume this should be Fig. 11b instead of Fig. 10b.

**Reviewer 3: Anonymous reviewer #3**

This paper presents a useful application -- the potential for near-real-time downscaling and bias-correction of mountain precipitation -- of a previously developed data assimilation method for snow water equivalent estimation. I found the paper well written and the application useful and compelling, and offer only a few very minor comments/errata for consideration by the authors.

1. l 76: 'plays' should be 'play' (the subject is 'basins')

We propose to include this edit (and other technical/grammatical edits proposed here) in the next draft of the manuscript.

2. l. 92: the verb should be are, not is (subject is 'Details')

We propose to accept this correction.

3. l. 102: SSiB-SAST should be defined, and the acronym LSM should be introduced on line 101.

We propose the following edit:

"First, an ensemble of prior snow estimates is generated using a forward land surface model (LSM); here, the modelling core is the SSiB-SAST LSM (Simplified Simple Biosphere - Snow-Atmosphere-Soil Transfer; Sun and Xue, 2001; Xue et al., 2003) paired with the Liston Snow Depletion Curve (Liston 2004)."

4. l. 115: consider 'ensemble member' rather than 'ensemble number'

We propose to accept this correction.

5. l. 125: My most significant comment -- Because of its coarse resolution (relative to the complex terrain of the Western US), MERRA-2 (or indeed any long-term global atmospheric reanalysis dataset) precipitation is likely fairly far from truth (e.g., Sun et al. 2017). The Bayesian framework can correct for quite a lot of precipitation error with its large(/very small) multipliers, but I'd like to see some comment/discussion about the potential impact of precipitation dataset quality on the framework described by the paper.

Testing different precipitation datasets in this framework (i.e., HRRR, ERA5, etc.) can be a pathway for future work. The greatest impact of this precipitation bias correction method will be in regions and with precipitation datasets that the reanalysis framework (i.e.: assimilation of fSCA observations) can effectively improve. MERRA2 was used in the original reanalysis (Fang et al., 2022) because it is globally available. The framework presented herein could be used with any precipitation dataset as the input and would provide product-specific corrections. Alternatively, the bias-corrected MERRA2 and prior MERRA2 could be used to derive bias-correction factors of other products. We propose to include additional text to describe these ideas in the Conclusions section in our response to reviewer 1's comment #2 and #15.

6. l. 290: 'eliminated' seems too strong of a word here -- I suggest 'greatly reduced' or something along those lines

We propose the following edit:

"This geographical distinction is lessened in Case A and Case B in 2015 and 2016, and greatly reduced in 2017."

7. Figure 6: It's rather hard to distinguish the shades of grey on panel a (sugg. hatching rather than shades of grey)

We propose to use vertical lines indicating percentiles instead of shades of gray or hatching, as follows, and include the following text revision to the caption:

"The 30th and 70th percentile are indicated with vertical lines, and differentiate between low snow (below 30th percentile), normal snow, and high snow (above 70th percentile) years."

[Figure]

8. l. 365 onward -- it seems like Figure 8 has been mistakenly referred to as Fig 7 throughout this section and on line 474.

We propose to include this correction.